# Kir2.1-mediated membrane potential promotes nutrient acquisition and inflammation through regulation of nutrient transporters

Weiwei Yu[1,2,7], Zhen Wang[1,2,7], Xiafei Yu[3,7], Yonghui Zhao[4], Zili Xie[4], Kailian Zhang[1], Zhexu Chi[1], Sheng Chen[1], Ting Xu[1], Danlu Jiang[1], Xingchen Guo[4], Mobai Li[1], Jian Zhang[1], Hui Fang[1], Dehang Yang[1], Yuxian Guo[1], Xuyan Yang[5], Xue Zhang ◉ [6], Yingliang Wu[4], Wei Yang[3✉] & Di Wang ◉ [1,2✉]

Immunometabolism contributes to inflammation, but how activated macrophages acquire extracellular nutrients to fuel inflammation is largely unknown. Here, we show that the plasma membrane potential ($V_m$) of macrophages mediated by Kir2.1, an inwardly-rectifying $K^+$ channel, is an important determinant of nutrient acquisition and subsequent metabolic reprogramming promoting inflammation. In the absence of Kir2.1 activity, depolarized macrophage $V_m$ lead to a caloric restriction state by limiting nutrient uptake and concomitant adaptations in nutrient conservation inducing autophagy, AMPK (Adenosine 5'-monophosphate-activated protein kinase), and GCN2 (General control nonderepressible 2), which subsequently depletes epigenetic substrates feeding histone methylation at loci of a cluster of metabolism-responsive inflammatory genes, thereby suppressing their transcription. Kir2.1-mediated $V_m$ supports nutrient uptake by facilitating cell-surface retention of nutrient transporters such as 4F2hc and GLUT1 by its modulation of plasma membrane phospholipid dynamics. Pharmacological targeting of Kir2.1 alleviated inflammation triggered by LPS or bacterial infection in a sepsis model and sterile inflammation in human samples. These findings identify an ionic control of macrophage activation and advance our understanding of the immunomodulatory properties of $V_m$ that links nutrient inputs to inflammatory diseases.

[1] Institute of Immunology and Sir Run Run Shaw Hospital, Zhejiang University School of Medicine, Hangzhou 310058, P. R. China. [2] Liangzhu Laboratory, Zhejiang University Medical Center, 1369 West Wenyi Road, Hangzhou, P. R. China. [3] Department of Biophysics, and Department of Neurology of the Fourth Affiliated Hospital, Zhejiang University School of Medicine, Hangzhou 310058, P. R. China. [4] State Key Laboratory of Virology, College of Life Sciences, Wuhan University, Wuhan 430072, P. R. China. [5] Department of Rheumatology of the Second Affiliated Hospital, Zhejiang University School of Medicine, Hangzhou 310058, P. R. China. [6] Department of Pathology and Pathophysiology, Zhejiang University School of Medicine, Hangzhou 310058, P. R. China. [7] These authors contributed equally: Weiwei Yu, Zhen Wang, Xiafei Yu. ✉email: yangwei@zju.edu.cn; diwang@zju.edu.cn

The activation of immune cells is intimately linked and dependent on profound and dynamic changes in cellular metabolism for diverse immune functions. Although the utilization of particular metabolic pathways is controlled by an exquisite balance of internal metabolites, reducing and oxidizing substrates, and mitochondrial adaptations, each immune cell metabolic phenotype is ultimately dependent on the acquisition of appropriate nutrients. Despite numerous studies examining the pathways and metabolites used by immune cells for host defense, little attention has been paid to the key question of how immune cells acquire extracellular nutrients to feed the metabolic underpinnings of inflammation.

Precise regulation of macrophage activation is essential for the control of inflammatory disease. Bacterial stimuli-activated inflammatory macrophages typically feature enhanced glycolytic metabolism, which not only produces ATP as energy to sustain their high secretory and phagocytic functions but also feeds glycolytic offshoots to drive inflammatory gene expression[1,2]. We and others have previously identified adequate supplies of essential amino acids such as serine and methionine as being critically involved in the induction of inflammatory cytokines including interleukin-1 (IL-1)[3,4]. To acquire constant extracellular nutrients, rapidly proliferating cells can upregulate glucose and glutamine transporter gene expression as well as prevent their lysosomal degradation[5]. However, macrophages are not prone to rapid proliferation upon inflammatory activation, and the strategies they adopt to ensure nutrient acquisition to fuel inflammation are poorly investigated.

The negative plasma membrane (PM) potential ($V_m$) generated by different ionic gradients across the membrane is a key electrochemical aspect of cells. Precise control of $V_m$ is essential for the physiological functions of excitable cells, such as neurons and cardiomyocytes. Of note, $V_m$ has recently been linked to cell survival and proliferation triggered by mitogenic signaling in excitable neuroblastoma cells and *Drosophila* neurons[6], suggesting broader functions of this electrochemical characteristic. Particularly, as non-excitable and less proliferative cells, the functions of $V_m$ in macrophages upon inflammatory stimuli are poorly recognized.

In this study, we show that macrophage $V_m$ mediated by the inwardly rectifying $K^+$ channel Kir2.1 contributes to metabolic and epigenetic programs of inflammation by ensuring nutrient acquisition. Mechanistically, this Kir2.1 regulation of immunometabolism relies on its control of the surface retention of key nutrient transporters via a modulation of PM phospholipid dynamics. This study illustrates an ionic control of Kir2.1 in inflammation by stabilizing macrophage $V_m$, and provides a potential approach to anti-inflammatory therapy.

## Results

**Macrophage PM depolarization limits nutrient acquisition required for inflammation.** To explore the impact of $V_m$ on inflammatory macrophages, we manipulated the $V_m$ of lipopolysaccharide (LPS)-stimulated peritoneal macrophages, measured by whole-cell patch clamping or the $V_m$ indicator DiBAC4(3)[7], by increasing extracellular $K^+$ ($[K^+]_e$) or applying gramicidin (Fig. 1a, Supplementary Fig. 1A, B), two widely used methods for plasma membrane (PM) depolarization but through distinct mechanisms[7–11]. Either method was used when the viability of LPS-stimulated macrophages was minimally affected by measuring both apoptosis and lytic cell death (Supplementary Fig. 1C, D). To seek the bona fide changes induced by PM depolarization, we performed combined analysis with RNA-sequencing (RNA-seq) data from depolarized LPS-stimulated macrophages treated with either elevated $[K^+]_e$ or gramicidin, and showed a strong

correlation between both treatments in terms of their impact on the transcriptome (Supplementary Fig. 1E). When we analyzed the shared differentially expressed genes between the two treatments by pathway enrichment analysis (Fig. 1b), the common downregulated genes were particularly enriched in 'inflammatory response' and 'defense response' (Fig. 1c), indicating an anti-inflammatory effect of PM depolarization in activated macrophages. By contrast, despite adequate extracellular nutrient resources, the common upregulated genes were mostly enriched in pathways typically induced under nutrient restriction, including 'response to starvation', 'autophagy', and 'catabolic process' (Fig. 1c). Consistently, gene set enrichment analysis (GSEA) showed significantly enrichment in 'autophagy', 'response to starvation', 'response of GCN2 to amino acid deficiency', and 'amino acid transport' in the settings of elevated $[K^+]_e$ and gramicidin (Fig. 1d, e). To determine whether these changed transcriptional programs were indeed attributable to a state of nutrient restriction, we performed unbiased metabolomics profiling with these depolarized LPS-stimulated macrophages for a similar combined analysis (Supplementary Fig. 1F). Although elevated $[K^+]_e$ and gramicidin showed some discrepancies in changed metabolites, possibly due to their mechanistic difference in depolarizing $V_m$, we found a substantial proportion of metabolites similarly altered by these two strategies (Fig. 1f). Using metabolic pathway enrichment analysis, the common decreased metabolites between elevated $[K^+]_e$ and gramicidin were mainly metabolites associated with glycolysis, pentose phosphate pathway, citric acid cycle, purine synthesis, phospholipid biosynthesis, and amino acid metabolism and biosynthesis (Fig. 1g–i and Supplementary Fig. 1G). Together with the upregulation of starvation-associated transcripts, these changes in metabolites suggested an impaired nutrient acquisition and subsequent anabolic metabolism upon PM depolarization. In sum, these data showed a profound impact of PM depolarization on the immunometabolism of macrophages upon inflammatory stimulation.

**Kir2.1 is a critical regulator of macrophage $V_m$ driving inflammation.** We next sought to identify a physiological modulator controlling this $V_m$-dependent regulation of inflammatory macrophages. Given the key roles of $K^+$ conductance in governing resting $V_m$[12], we compared the expression of 54 annotated $K^+$ channel genes in bone marrow-derived macrophages (BMDMs) using RNA-seq data, and found several highly expressed $K^+$ channels including $K_{Ca}3.1$, THIK1, TWIK2, Kir2.1, and $K_v1.3$ (encoded by *Kcnn4*, *Kcnk13*, *Kcnk6*, *Kcnj2*, and *Kcna3*) (Fig. 2a). Among them, we first excluded the contributions of $K_{Ca}3.1$ and $K_v1.3$ to macrophage $V_m$, because their selective inhibitors TRAM-34 and PAP-1 had no effect on the $V_m$ of LPS-stimulated macrophages as compared to elevated $[K^+]_e$ or gramicidin (Supplementary Fig. 2A). THIK1 and TWIK2 belong to the two-pore domain $K^+$ channel ($K_{2P}$) family giving rise to background $K^+$ currents responsible for stabilizing resting $V_m$[13], which have been identified as major $K^+$ efflux channels triggering NLRP3 inflammasome-induced inflammation in macrophages and microglia[14,15]. Given these studies on $K_{2P}$ channels in macrophages and a high compensatory effect among different $K_{2P}$ channels[13], we focused on the inwardly rectifying $K^+$ channel Kir2.1, which is of vital importance in regulating membrane excitability as a critical regulator of resting $V_m$ in excitable cells such as cardiomyocytes and smooth muscle cells[16–18], while knowledge of its functions in non-excitable cells, like immune cells in particular, remains ill defined.

To examine the electrophysiological properties of macrophage Kir2.1, we made whole-cell patch-clamp recordings using peritoneal macrophages treated with ML133, a widely used

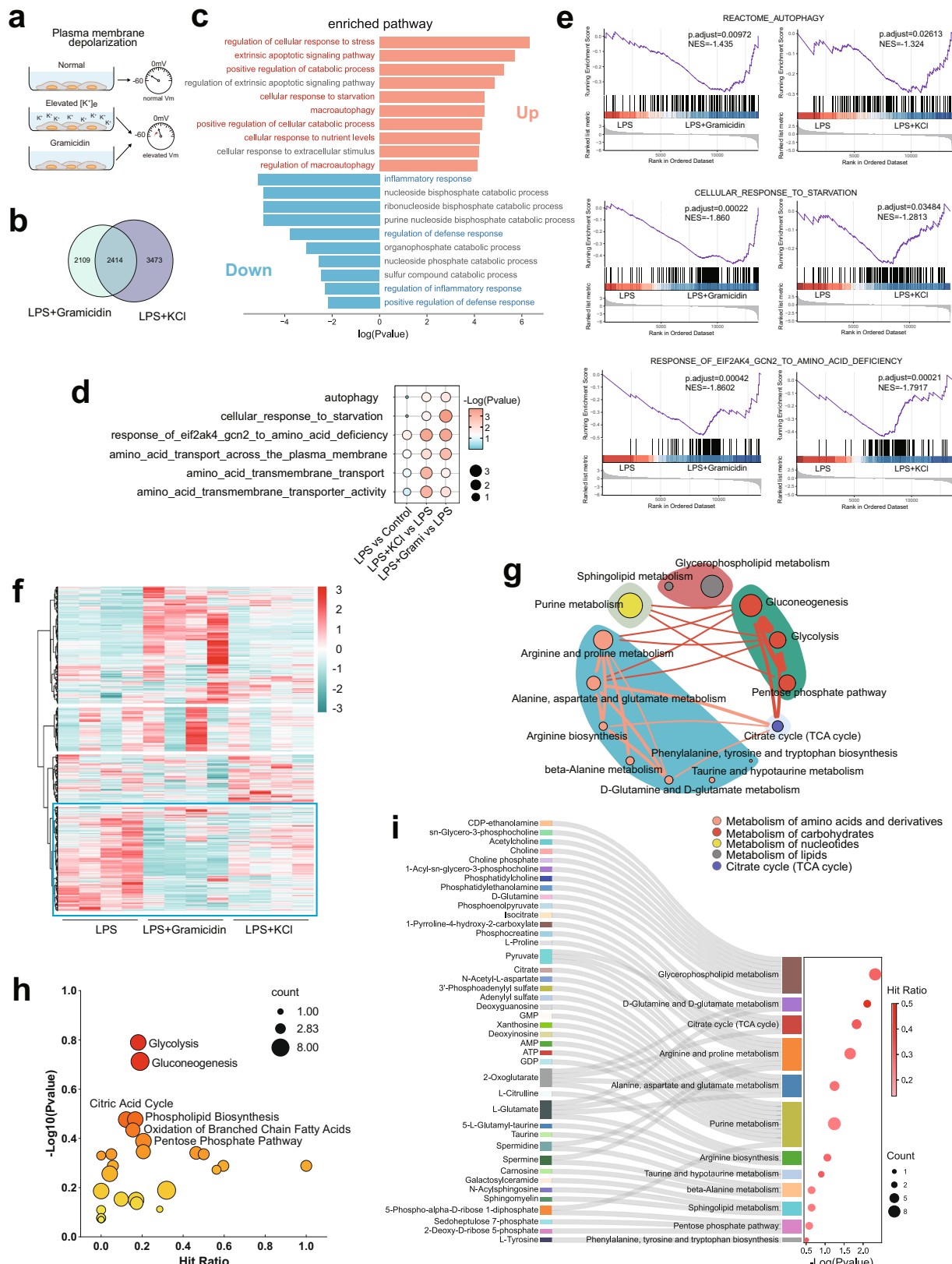

selective Kir2 blocker with an IC$_{50}$ of 1.9 μM for Kir2.1[19,20], or peritoneal macrophages from *Lyz2-cre-Kcnj2$^{f/f}$* mice with a specific deletion of Kir2.1 in myeloid cells (Supplementary Fig. 2B). A typical Kir2.1 current[17,18,21–23] was recorded, and ML133 potently inhibited these Kir2.1 currents in both resting (Fig. 2b, e) and LPS-stimulated macrophages (Fig. 2c, e) without

affecting cell viability (Supplementary Fig. 2C). Consistent with this, the Kir2.1 current was also abolished in Kir2.1-deficient macrophages (Fig. 2d, e), while the ability of bone marrow cells to differentiate into macrophages was unaffected (Supplementary Fig. 2D, E). Most important, we found a depolarized V$_m$ in both Kir2.1-deficient resting and activated macrophages (Fig. 2f and

**Fig. 1 Macrophage $V_m$ is essential for nutrient acquisition to fuel inflammation. a** Strategy for depolarizing macrophages and membrane potential measurement. **b, c** Overlap analysis of differentially expressed genes in mouse BMDMs treated with 500 ng/ml LPS in the presence or absence of elevated $[K^+]_e$ (50 mM) or gramicidin (1.25 μM) for 6 h. The pathway enrichment analysis of overlapped downregulated (reference threshold: fold-change, <0.7; $p$ value, <0.05) and upregulated (reference threshold: fold-change, >1.3; $p$ value, <0.05) genes were performed in Metascape. **d, e** GSEA of mouse BMDMs treated with 500 ng/ml LPS in the presence or absence of elevated $[K^+]_e$ (50 mM) or gramicidin (1.25 μM) for 6 h. **f** Heatmap of metabolites identified in unbiased metabolomics. ($n = 4$). Data analyzed by RStudio and the row clustering distance and clustering method were "euclidean" and "ward.D2", respectively. The highlighted cluster represents metabolites downregulated in both the elevated $[K^+]_e$ and gramicidin groups *versus* the LPS group. **g–i** Pathway Enrichment analysis of metabolites in the highlighted cluster with MetaboAnalyst 5.0 based on the KEGG database (**h**), SMPDB (The Small Molecule Pathway Database) database (**i**), and the combined KEGG and SMPDB database analysis (**g**). The enriched metabolic pathways are categorized into different pathway groups based on the Reactome metabolic pathway database. The size of circle for each pathway represent counts of enriched metabolites. The thickness of lines between pathways represents counts of common metabolites between groups (**g**). Source data are provided as a Source Data file.

Supplementary Fig. 2F). ML133 also depolarized $V_m$ in a dose-dependent manner and 25 μM ML133 led to a $V_m$ depolarization (from $-42.8 \pm 2.9$ mV to $-30.6 \pm 1.3$ mV) comparable to that induced by Kir2.1 deficiency (from $-42.8 \pm 2.9$ mV to $-29.7 \pm 2.2$ mV) (Fig. 2f), which led us to choose the dose of 25 μM ML133 in parallel with Kir2.1 deficiency in most of the subsequent experiments. No additional effect of ML133 in Kir2.1-deficient macrophages further confirmed its specificity (Fig. 2e, f). Moreover, the effect of ML133 on macrophage $V_m$ persisted throughout the early stage of LPS stimulation (Supplementary Fig. 2G).

The major function of macrophages in response to bacterial stimuli is to elicit inflammation for host defense. GSEA of RNA-seq data from LPS-stimulated BMDMs demonstrated a striking enrichment of 'inflammatory response' after ML133 treatment (Fig. 2g), recapitulating the anti-inflammatory effect of PM depolarization in macrophages. Further analysis showed that a cluster of inflammatory genes including *Il1b*, *Il1a*, *Il6*, *Il18*, *Il12a*, and *Cxcl10* were specifically suppressed by ML133, while another inflammatory gene, *Tnf*, was not affected (Fig. 2h). Consistent with this, we found a similar reduction of *Il1b* and *Il1a* mRNA as well as pro-IL-1β protein expression by either ML133 or Kir2.1 deficiency in LPS-stimulated peritoneal macrophages (Fig. 2i, Supplementary Fig. 2H, I), while no change was found in *Tnf* (Fig. 2i). The absence of additional effects of ML133 in Kir2.1-deficient macrophages (Fig. 2i) and the similar results obtained by another Kir2 inhibitor, pentamidine analogue 6 (PA-6)[24–26] (Supplementary Fig. 2J), further confirmed the specificity of Kir2.1. We next sought to determine whether these inflammatory genes suppressed by ML133 were also responsive to PM depolarization by other means. Remarkably, elevated $[K^+]_e$ or gramicidin showed a similar suppression of these genes, while inhibition of $K_{Ca}3.1$ or $K_v1.3$ did not (Fig. 2j), concordant with their minor effect on $V_m$ (Supplementary Fig. 2A). Moreover, these effects were not due to an osmotic effect, as choline chloride or mannitol did not induce a similar suppression of IL-1β production (Supplementary Fig. 2K). These results indicate that Kir2.1 is a physiological regulator of macrophage $V_m$ and stimulates inflammatory macrophages in vitro.

**Kir2.1-mediated $V_m$ is essential for nutrient acquisition endowing macrophages with the metabolic signature of inflammation.** Given that the production of TNF was unaffected, we hypothesized that Kir2.1 would not be required for the general signaling pathways essential for most LPS-induced genes. Indeed, ML133 or Kir2.1 deficiency had little effect on LPS-induced NF-κB and MAPK activation (Supplementary Fig. 3A, B). We next considered the possibility that Kir2.1 promotes inflammation by modulating $Ca^{2+}$-dependent signaling pathways, because $K^+$-mediated $V_m$ could indirectly affect $Ca^{2+}$ flux[12]. However,

the ML133-induced IL-1β suppression was independent of the concentrations of extracellular $Ca^{2+}$ or the other divalent cations $Mg^{2+}$ and $Mn^{2+}$ (Supplementary Fig. 3C, D), as well as extracellular $Ca^{2+}$ chelation by BAPTA or EGTA (Supplementary Fig. 3E–G). In the setting of chelating intracellular $Ca^{2+}$ by BAPTA-AM, we found that this treatment alone was sufficient of suppress IL-1β and IL-1α (Supplementary Fig. 3H), and had an additional effect on the basis of ML133 (Supplementary Fig. 3I). Moreover, when we depleted intracellular $Ca^{2+}$ stores using thapsigargin (TG), ML133 showed a similar inhibitory efficiency on IL-1β production in the presence or absence of TG (Supplementary Fig. 3J), suggesting a separate mechanism of Kir2.1 in parallel to that of intracellular $Ca^{2+}$.

Previous studies have reported a synergism of glycolytic offshoots and amino acid metabolism driving the expression of a cluster of inflammatory genes, including *Il1b*, *Il1a*, *Il6*, *Il18*, *Il12a*, *Il27*, and *Cxcl10*, but not *Tnf*[3,4]. Surprisingly, the suppression of such a cluster of metabolism-responsive inflammatory genes by limiting nutrient availability was recapitulated by ML133 (Fig. 2h), suggesting a metabolic mechanism underlying the regulation of Kir2.1. Consistent with this idea, unbiased metabolomic profiling of LPS-stimulated macrophages showed decreased metabolites representing anabolic metabolism after ML133 treatment (Fig. 3a, b) similar to the results in the setting of PM depolarization with elevated $[K^+]_e$ or gramicidin (Fig. 1f–i). Both unbiased metabolomics profiling and a targeted metabolomics approach showed a significant reduction of LPS-induced glycolysis intermediates by ML133 (Fig. 3c and Supplementary Fig. 3K), which was further validated by the decreased extracellular acidification rate (ECAR) upon ML133 treatment or Kir2.1 deficiency (Fig. 3d). The metabolites R5P and 3PS, representing the glycolytic offshoots pentose phosphate pathway (PPP) and serine synthesis pathway (SSP), respectively (Fig. 3c), as well as the amino acids essential for metabolism-responsive inflammatory genes[3,4] (Fig. 3e), were also decreased by ML133. In addition, these metabolic alterations were attributable to a reduction in electrochemically dependent nutrient uptake, because macrophages with depolarized $V_m$ due to ML133 or Kir2.1 deficiency had diminished LPS-induced glucose uptake, which was recapitulated by gramicidin-induced PM depolarization (Fig. 3f). Similar results were obtained in *Lyz2-cre-Kcnj2^{f/f}* mice by measuring the LPS-induced glucose uptake of peritoneal exudate cells (PECs) and inflammatory monocytes in vivo (Fig. 3g). In addition, by assessing $^{13}C$-labeled serine (m + 3) and glycine (m + 2) derived from U-[$^{13}C$]-glucose tracers, the SSP fueled by glucose uptake was concomitantly impaired upon PM depolarization by both Kir2.1 repression (ML133 treatment or Kir2.1 deficiency) (Fig. 3h, i) and electrochemical means (elevated $[K^+]_e$ or gramicidin) (Fig. 3j). In addition, unlabeled (m + 0) serine, glycine, and methionine were similarly decreased in these activated macrophages with PM depolarization

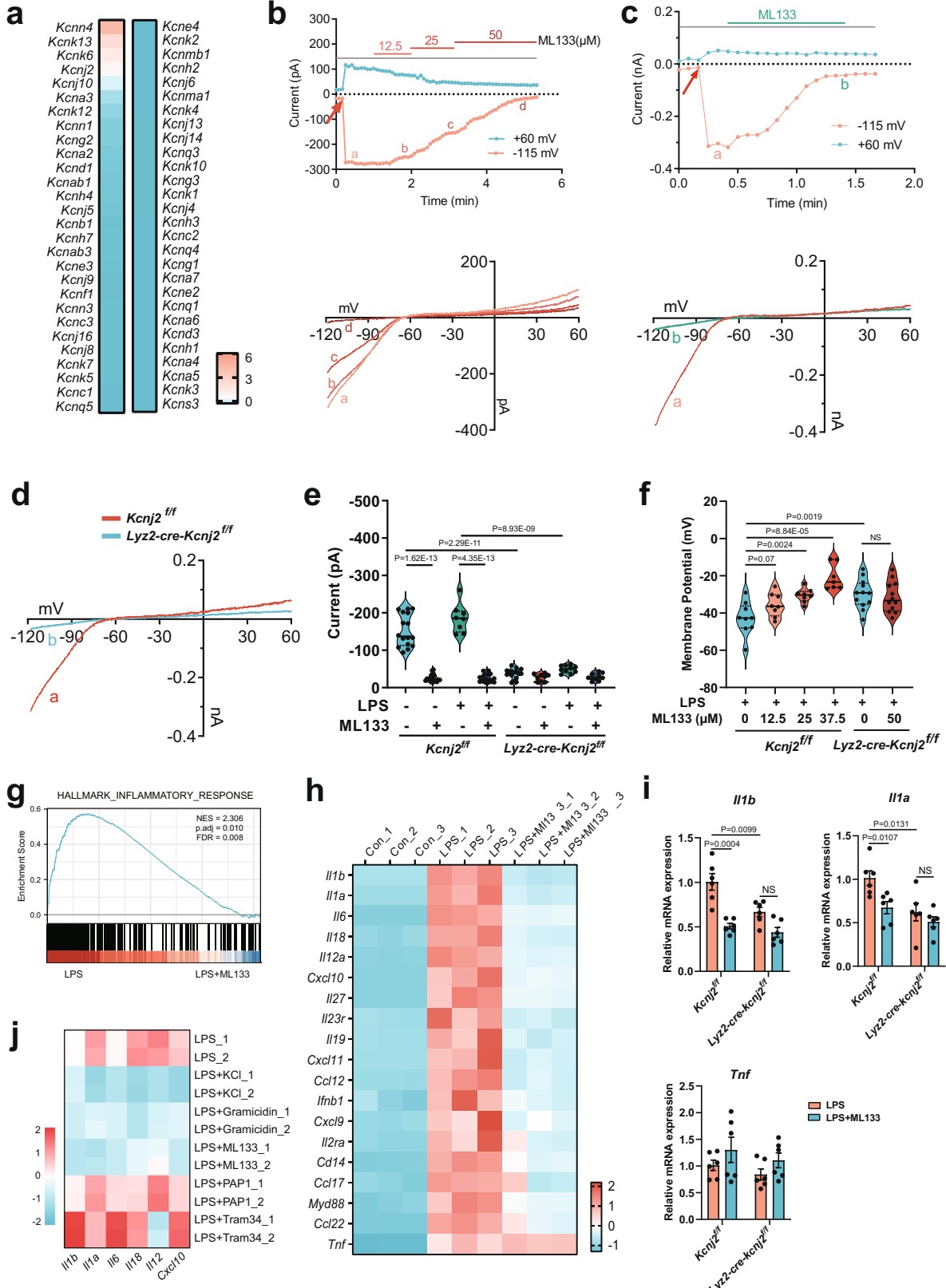

(Fig. 3h–j). These data together indicate a profound impact of Kir2.1-mediated $V_m$ on nutrient acquisition of macrophages upon activation. Consistent with the changes in metabolites, RNA-seq analysis showed that the enzymes mediating the three-step SSP—*Phgdh*, *Psat1*, and *Psph*, as well as positive regulators of the SSP[27], were all upregulated by ML133 (Supplementary

Fig. 3L), showing a similar phenotype upon serine starvation[28,29]. GSEA also highlighted significant enrichment in 'amino acid transport' and 'serine, glycine, one-carbon (SGOC) metabolism' (Fig. 3k). We further analyzed the differentially expressed genes shared between ML133 treatment and Kir2.1 deficiency by pathway enrichment analysis. Besides the common

**Fig. 2 Kir2.1 is a critical regulator of macrophage $V_m$ driving inflammation. a** Expression profile of $K^+$ channels in mouse bone marrow-derived macrophages (BMDMs) analyzed by RNA-seq ($n = 3$). The FPKM data of different genes are standardized to the Z-score by column in IBM SPSS Statistics. **b–d** Electrophysiological properties of freshly isolated WT (**b–d**) and Kir2.1-knockout (**d**) mouse peritoneal macrophages using patch-clamp. Thioglycolate-elicited mouse peritoneal macrophages were treated with (**b**) or without (**c**) 500 ng/ml LPS for 1 h before recording. The time-dependent current traces were recorded at a clamp voltage of –115 mV or +60 mV (red arrow, initiation of recording). The lower panel shows the I–V curves of the corresponding points for different doses of ML133 in the left panel. **d** Representative I–V curves of WT and Kir2.1-knockout mouse peritoneal macrophages. **e** Statistics of current amplitude recorded at a clamp voltage of –115 mV. Macrophages were treated and clamped as described above ($Kcnj2^{f/f}$, $n = 16$, 18, 8, and 17 respectively; $Lyz2$-cre-$Kcnj2^{f/f}$, n = 16, 9, 10, and 10 respectively; mean ± SEM). **f** Statistics of Vm recorded at 3 min after initiation of recording ($Kcnj2^{f/f}$, n = 10, 7, 8, 9, 9, 8, 8, and 7, respectively; $Lyz2$-cre-$Kcnj2^{f/f}$, n = 9, 8, 12, and 12, respectively; mean ± SEM). **g, h** RNA-seq analysis of inflammatory genes. GSEA analysis showing enrichment of the inflammatory response genes (**g**) and heatmap of inflammatory genes (**h**). NES, normalized enrichment score; FDR, false discovery rate. ($n = 3$). **i** $Il1a$, $Il1b$, and $Tnf$ mRNA transcription assays. Mouse peritoneal macrophages from WT or $Lyz2$-cre-$Kcnj2^{f/f}$ mice treated with 500 ng/ml LPS for 6 h in the presence or absence of ML133 (25 μM) followed by qPCR analysis ($n = 6$, mean ± SEM). **j** Mouse peritoneal macrophages treated with or without 500 ng/ml LPS in the presence or absence of elevated $[K^+]_e$ (50 mM), gramicidin (1.25 μM), ML133 (25 μM), PAP1 (Kv1.3 inhibitor, 1 μM), Tram34 (KCa3.1 inhibitor, 1 μM) for 6 h followed by qPCR analysis of mRNA transcription. Two-tailed unpaired Student's t-test. Source data are provided as a Source Data file.

downregulated genes mostly enriched in 'inflammatory response' and 'response to LPS', well reflecting the anti-inflammatory effect of Kir2.1 repression (Fig. 3l), the common upregulated genes were found to be particularly enriched in pathways 'in response to starvation' (Fig. 3m). Among these genes, we noted the master sensors and regulators in response to nutrient starvation including GCN2, PERK, IMPACT, and SLC38A2[30] (Supplementary Fig. 3M), and confirmed the activation of GCN2 after ML133 treatment (Supplementary Fig. 3N). These results together highlight a significant importance of Kir2.1 in nutrient acquisition to feed the metabolic underpinnings of inflammation.

**Kir2.1-mediated $V_m$ supports nutrient uptake for SAM generation to induce a metabolism-responsive inflammatory program.** The impaired PPP and SSP fueled by glucose uptake and decreased serine and methionine uptake led us to quantify S-adenosylmethionine (SAM), whose synthesis is synergistically fed by these metabolic inputs (Fig. 4a)[4,27,31]. As the primary methyl donor, SAM availability directly modulates epigenetic methylation marks intimately linked to the chromatin state and gene transcription[32–36]. Either ML133 treatment or Kir2.1 deficiency led to a significant reduction in the cellular SAM of LPS-stimulated macrophages (Fig. 4b, c). To determine whether this effect was due to a lack of building blocks fueled by upstream nutrient inputs, we used U-[$^{13}$C]-glucose tracers and measured a reduction of LPS-induced m + 5 to 9 SAM (via both the PPP and SSP) as well as their precursors of m + 5 to 9 ATP (Fig. 4d and Supplementary Fig. 4A) in the settings of PM polarization with ML133, elevated $[K^+]_e$, and gramicidin (Fig. 4e, f). We next examined the SAM generation fueled by exogenous amino acid uptake (Fig. 4g and Supplementary Fig. 4B) and found a similar reduction of m + 1 to 4 SAM fed with U-[$^{13}$C]-serine-derived carbons (Fig. 4h). Although these results provided a correlation between the decrease in this critical epigenetic modulator and the depression of a cluster of metabolism-responsive inflammatory genes upon PM depolarization, we sought to determine whether this association was causal. To this end, we supplemented ML133-treated inflammatory macrophages with exogenous SAM and found that the suppressed $Il1b$ mRNA was dose-dependently restored (Fig. 4i), as well as the protein expression of pro-IL-1β (Supplementary Fig. 4C). The limited nutrient uptake and subsequent caloric restriction also led us to quantify acetyl-coenzyme A (AcCoA), a molecule that enables carbons derived from diverse nutrients to be utilized in the citric acid cycle for oxidative phosphorylation, and also serves as a metabolic substrate for histone and non-histone protein acetylation[37]. Indeed, LPS-induced AcCoA was significantly decreased by ML133 (Supplementary Fig. 4D), a finding consistent with the nutrient starvation

state due to PM depolarization. Interestingly, when we supplemented ML133-treated inflammatory macrophages with exogenous acetate, the immediate precursor of AcCoA, IL-1β and IL-1α production was not restored (Supplementary Fig. 4E). Another important epigenetic metabolite, the citric acid cycle intermediate α-ketoglutarate (αKG)[37], also showed no rescue effect on IL-1β production (Supplementary Fig. 4F). These results suggested a greater dependence of the $V_m$-induced inflammatory program on the epigenetic programs fueled by SAM generation. To assess the epigenetic consequences of limited SAM availability, we analyzed the expression of 183 annotated SAM-dependent methyltransferase enzymes[38] and found a remarkable upregulation of all methyltransferases for histone H3 methylation at lysine 36 (H3K36me)[39] (Fig. 4j and Supplementary Fig. 4G) compared to those for other methylation marks (Supplementary Fig. 4H). Among them, ASH1L and NSD1 have specific mono- and dimethylase activity for H3K36, and SETD2 is the only reported trimethylase catalyzing the trimethylation of H3K36[39,40]. Consistent with the main distribution of H3K36me3 in a wide range of gene body regions as a SAM 'sink', making it more sensitive to SAM availability for gene expression[39,41], chromatin immunoprecipitation with quantitative PCR (ChIP-qPCR) showed that PM depolarization by Kir2.1 repression (Fig. 4k, l) or elevated $[K^+]_e$ (Fig. 4m) decreased LPS-induced H3K36me3 occupancy in $Il1b$ gene-body regions[42], as well as those of other metabolism-responsive genes including $Il1a$, $Il18$, and $Cxcl10$ (Fig. 4n). By contrast, H3K36me3 enrichment in the gene-body region of $Tnf$ was less affected (Supplementary Fig. 4I), concordant with its unchanged mRNA expression upon PM depolarization (Fig. 2h, i). Collectively, these data suggest that Kir2.1 repression leads to the depletion of intracellular SAM fueling histone methylation marks that at least include H3K36me3, thereby limiting the epigenetic imprints of metabolism-responsive inflammatory genes.

**Kir2.1-mediated $V_m$ drives nutrient uptake by retaining nutrient transporters on the macrophage cell surface.** Given the analogous phenotypes in the metabolic-epigenetic coordination and a subsequent metabolism-responsive inflammatory program under different PM depolarization conditions, we reasoned that there would be common changes in the gene transcripts representing a specific mechanism underlying the $V_m$ control of immunometabolism. Thus, we performed combined analysis with the RNA-seq data from LPS-stimulated macrophages in the presence of ML133, gramicidin, or elevated $[K^+]_e$, and reveled 1172 differentially expressed genes shared by all three treatments (Fig. 5a). Consistent with the above findings, the common downregulated genes were mostly enriched in the pathways of

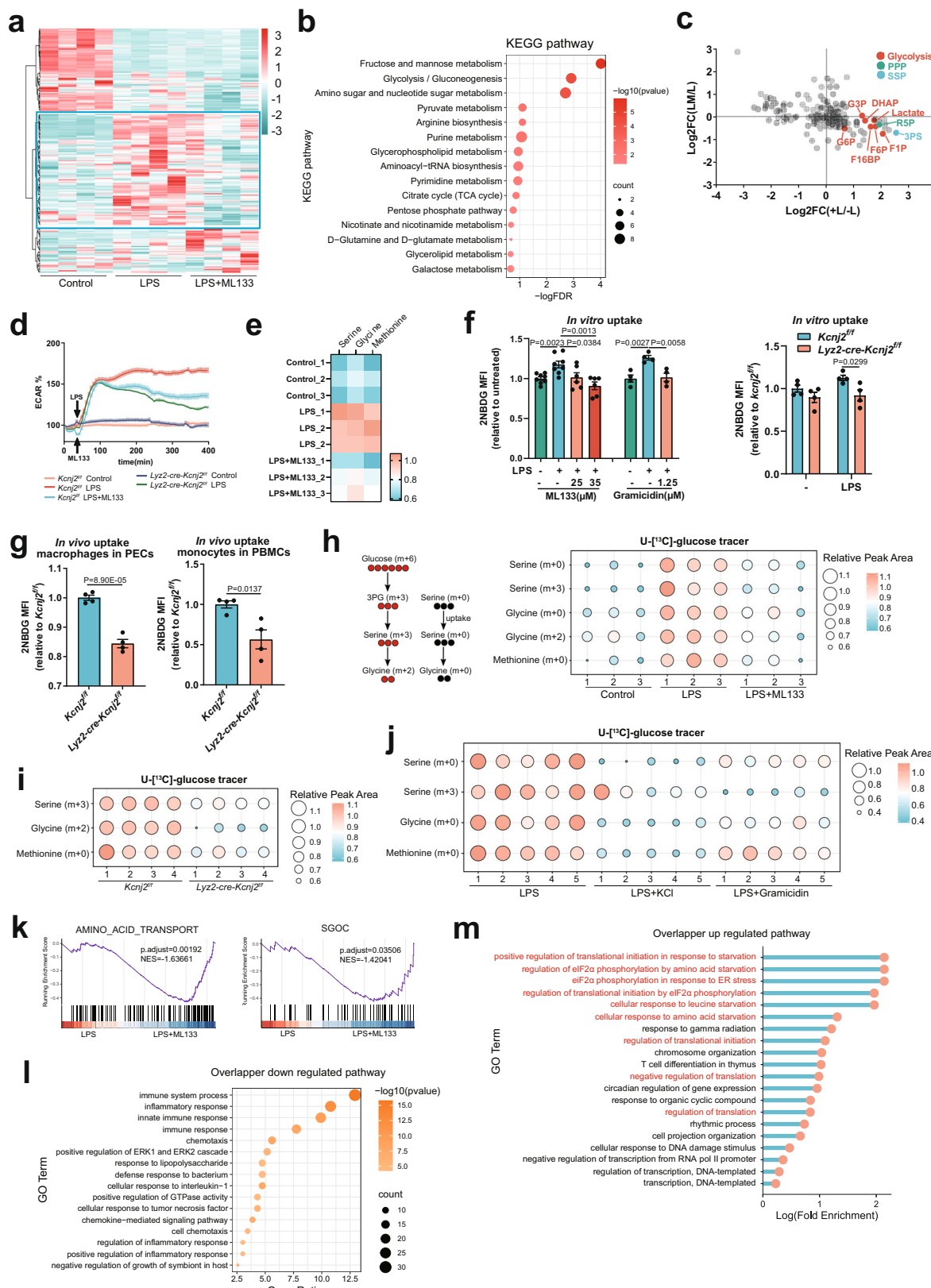

'inflammatory response' (Fig. 5b), and the common upregulated genes were significantly enriched in key pathways of metabolic adaptions in response to nutrient starvation such as 'autophagy', 'amino acid transport', and 'negative regulation of mTOR signaling' (Fig. 5c). The increased autophagy was confirmed by measuring the abundance of the autophagosome proteins LC3b-I and the PE-conjugated form LC3b-II (Fig. 5d). In addition, immunoblot analysis showed increased activity of the energy sensor AMPK (Fig. 5e), suggesting an induction of nutrient conservation upon PM depolarization[1,43,44]. Remarkably, we noted the upregulation of a set of solute carrier (SLC) family members, including the 4F2 heavy chain (4F2hc, also known as

**Fig. 3 Kir2.1-mediated $V_m$ endows macrophages with the metabolic signature of inflammation. a** Heatmap of metabolites identified by unbiased metabolomics in mouse peritoneal macrophages ($n = 4$). The second cluster represents metabolites upregulated in LPS-treated *versus* control groups and then downregulated in the in LPS plus ML133 group (highlighted). **b** KEGG pathway analysis of metabolites in the second cluster with MetaboAnalyst 5.0. **c** Metabolites of glycolysis, and the PPP and SSP pathways identified in unbiased metabolomics. L: LPS, LM: LPS + ML133. **d** Extracellular acidification rate (ECAR) analyzed by Seahorse. ($n = 6$, mean ± SD). **e** Quantification of total serine, glycine, and methionine in cells treated with 500 ng/ml LPS in the presence or absence of 25 μM ML133 for 6 h followed by LC−MS.($n = 3$). **f** In vitro glucose uptake assays of mouse peritoneal macrophages. ($n = 8, 8, 6, 6, 4, 4, 4$ respectively for left panel; $n = 4$ for right panel; mean ± SD). **g** In vivo glucose uptake assays. Glucose uptake by inflammatory macrophages (CD45[+] CD11b[+] F4/80[+]) in PECs and monocytes (CD45[+] CD11b[+] Ly6C[high]) in PBMCs was measured as 2NBDG MFI by FACS. ($n = 4$, mean ± SEM). **h, i** Mass isotopomer distribution analysis of metabolites derived from glucose. ($n = 3$ or $4$, respectively). **j** Mass isotopomer distribution analysis of metabolites derived from glucose in mouse peritoneal macrophages treated with 500 ng/ml LPS for 6 h in the presence of physiological or elevated $[K^+]_e$ (50 mM) or gramicidin (1.25 μM) in medium containing U-[$^{13}$C]-glucose (25 mM) ($n = 5$). **k**–**m** RNA-seq analysis of control and *Lyz2-cre-Kcnj2[f/f]* mouse BMDMs. GSEA analysis showing enrichment (**k**). The pathway enrichment analysis of overlapped downregulated (reference threshold: fold-change, <0.5; $p$ value, <0.05) (**l**) and upregulated (reference threshold: fold-change, >1.5; $p$ value, <0.05) (**m**) genes by both ML133 treatment and Kir2.1-deficiency in LPS-primed BMDMs. NES normalized enrichment score; ($n = 3$). Two-tailed unpaired Student's *t*-test. Glucose uptake data are representative of three independent experiments. Gating strategies for FACS were shown in Supplementary Figs. 8 and 9. Source data are provided as a Source Data file.

CD98 or SLC3A2), a critical player in nutrient uptake (Fig. 5c). 4F2hc acts as a chaperone that interacts with different light chains to form stable heterodimeric transporters for the uptake of various amino acids, as well as facilitating the stabilization of glucose transporter 1 (GLUT1), the primary glucose transporter in macrophages[5,45,46]. We thus considered whether Kir2.1-mediated $V_m$ supported nutrient acquisition to fuel inflammation by modulating nutrient transporters. Using flow cytometry, we detected an induction of surface expression of both 4F2hc and GLUT1 upon LPS stimulation, which was dose-dependently inhibited by ML133 (Fig. 5f), and similarly by PM depolarization with elevated $[K^+]_e$ or gramicidin (Fig. 5g, h). The decreased surface expression of GLUT1 was further confirmed by immunoblotting for biotinylated cell-surface proteins (Fig. 5i) in macrophages with abolished Kir2.1, elevated $[K^+]_e$, or gramicidin (Fig. 5j–m). Therefore, the impeded nutrient uptake of macrophages with a depolarized $V_m$ is possibly because of the loss of critical nutrient transporters on the cell surface.

**Kir2.1 facilitates cell surface retention of nutrient transporters by configuring plasma membrane phospholipid dynamics.** According to the combined analysis, we noted that the pathway of 'endocytosis' was also significantly enriched in the genes jointly affected by ML133, gramicidin, and elevated $[K^+]_e$ (Fig. 5b), providing a possible mechanistic link between PM depolarization and the loss of surface nutrient transporters. To gain a deeper understanding of the impact of PM depolarization on the molecular events occurring around the cell surface, we extracted the PM proteins of LPS-stimulated macrophages for further quantitative proteomics analysis (Fig. 6a). 215 differentially expressed proteins were shared between the treatments with ML133 and elevated $[K^+]_e$ (Supplementary Fig. 5A), and the pathways related to membrane trafficking including 'vesicle-mediated transport', 'endocytosis' and 'phagocytosis' were significantly enriched (Fig. 6b, c). These different omics data together suggest a contribution of endocytosis to the depolarized $V_m$-induced loss of cell surface nutrient transporters, which is consistent with the view that nutrient uptake is tightly controlled by the dynamic membrane trafficking of such transporters[5]. In addition, the membrane trafficking of nutrient transporters, including GLUT1 and 4F2hc, are usually co-regulated in response to different stresses, possibly because of their clustering in the same membrane domains[5,47,48]. We found that, after ML133 or elevated $[K^+]_e$ treatment, LPS-stimulated macrophages showed significantly enhanced internalization of both GLUT1 and 4F2hc (Fig. 6d). A substantial part of glucose transporter internalization has been reported to be blocked by the endocytosis inhibitor nystatin[49,50].

Without affecting LPS/TLR4-mediated downstream signaling (Supplementary Fig. 5B), nystatin partly restored the decreased levels of surface GLUT1 (Fig. 6e, f), glucose uptake (Fig. 6g), and IL-1β production (Fig. 6h) upon Kir2.1 repression. Consistent with this, the enrichment of H3K36me3 in the gene-body regions of *Il1b, Il1a, Il18*, and *Cxcl10* loci in Kir2.1-depleted macrophages was also restored (Supplementary Fig. 5C). GLUT1 and 4F2hc are ARF6/GRP1 cargo proteins that can be recycled back to the plasma membrane via the tubular recycling endosome[5,47,51,52]. Strikingly, a constitutively active mutant of GRP1 (S155D/T280D, GRP1 DD mutant), which forces the recycling of these transporters back to the PM and prevents their loss[51,53], largely restored the ML133-induced suppression of IL-1β (Fig. 6i), further indicating the $V_m$ regulation of inflammation by preventing the loss of surface nutrient transporters.

Essentially, a depolarized $V_m$ leads to changes in the electrochemical properties of the PM, which is composed mainly of charged phospholipids. PM depolarization has recently been recognized to affect PM phospholipid dynamics, particularly the distribution of phosphatidylinositol 4,5-bisphosphate (PIP$_2$)[6], whose enrichment in PM microdomains is of considerable importance in membrane trafficking and endocytosis by triggering PM invaginations and recruiting endocytic factors[54–58]. Although the amount of PM PIP$_2$ was unchanged (Supplementary Fig. 5D, E), ML133 treatment or elevated $[K^+]_e$ led to significantly enhanced PIP$_2$ clustering on the PM of LPS-stimulated macrophages by immunofluorescent staining with anti-PIP$_2$ antibody[59] and three-dimensional imaging (Fig. 6j–l). More importantly, the ML133-induced decreases in glucose uptake and IL-1β production were significantly restored by the expression of lipid phosphatase SopB (Fig. 6m), an effective means of inhibiting endocytic trafficking by depleting the membrane level of PIP$_2$[60–62]. Collectively, these data suggest a critical regulation by which Kir2.1-mediated $V_m$ modulates membrane phospholipid dynamics for the surface retention of nutrient transporters, thus ensuring nutrient acquisition to fuel inflammation.

**Kir2.1 repression alleviates LPS- and bacterial infection-induced inflammation in vivo and pathological inflammation in human samples.** Last, to explore the physiological relevance of this regulation, we examined the phenotype of inflammation in vivo when the Kir2.1 regulation of $V_m$ was absent. Using an in vivo LPS-induced sepsis model, we measured decreased serum levels of IL-1β, IL-1α, and IL-6 as well as increased survival of mice treated with ML133 (Fig. 7a–c) or in *Lyz2-cre-Kcnj2[f/f]* mice (Fig. 7d, e) compared to controls. This effect was similarly found in bacterial infection-induced inflammation, as ML133 or Kir2.1

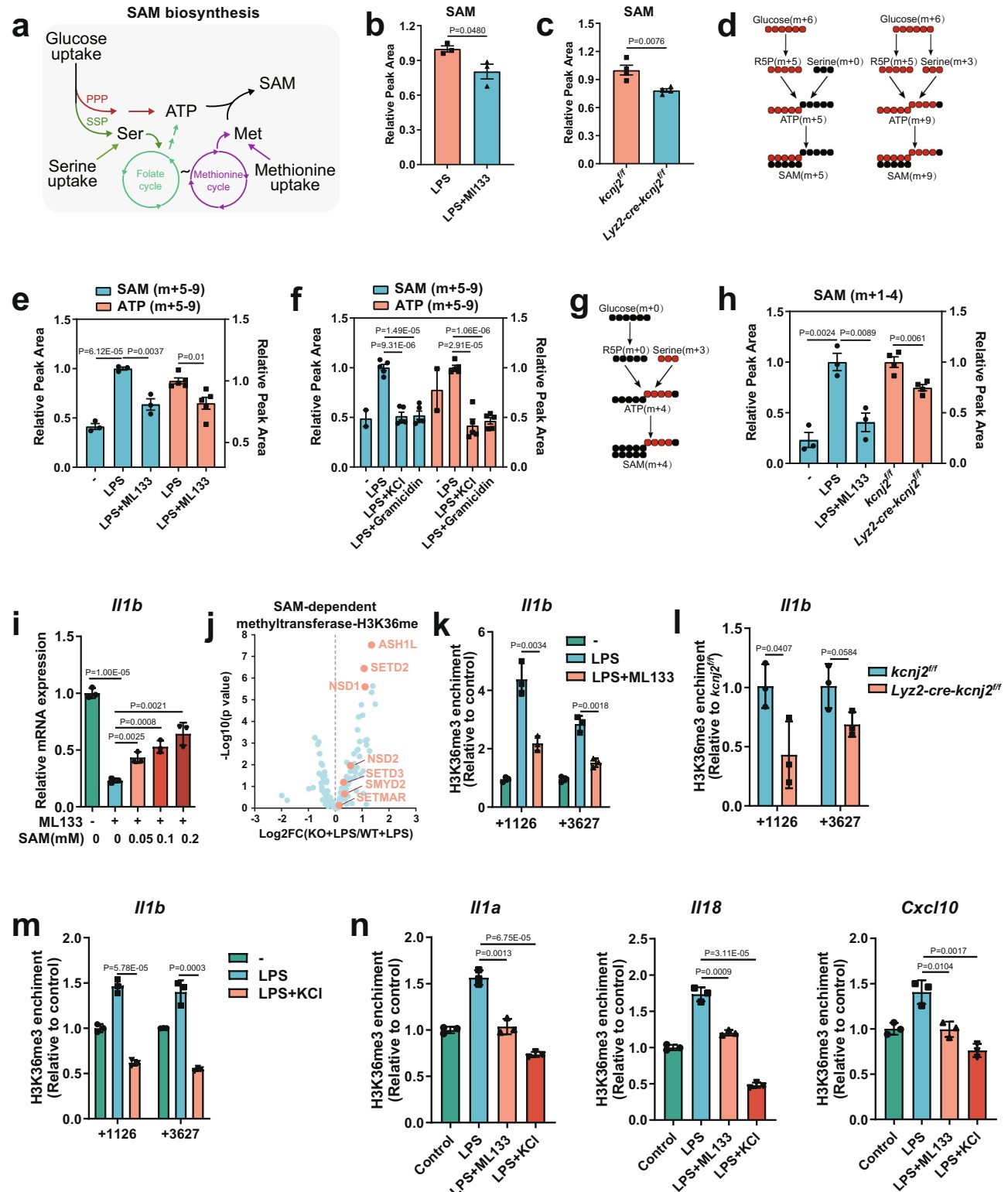

deficiency led to a reduction in IL-1β and IL-1α in response to the infection with gram-negative bacteria *Escherichia coli* and *Salmonella typhimurium* (strain SL1344) both in vitro (Supplementary Fig. 6A, B) and in vivo (Fig. 7f). In human samples, previously generated transcriptomic data from the blood monocytes of patients during sepsis and after recovery[63] showed that *KCNJ2* was remarkably upregulated in monocytes from sepsis patients compared to healthy donors (Fig. 7g). In addition, *KCNJ2* was more strongly increased by LPS stimulation in

monocytes from healthy and recovering donors (an endotoxin-sensitive phenotype) than that in monocytes from septic donors (an endotoxin-tolerant phenotype), with similar patterns of the cluster of metabolism-responsive inflammatory genes (Fig. 7g, h). These data suggested an association of Kir2.1-mediated inflammation with sepsis in humans. To further strengthen the evidence for the role of Kir2.1 in driving inflammation in human samples, we used synovial fluid cells from gouty patients, an inflammatory arthritis sensitive to anti-IL-1 treatment[64]. Strikingly, ML133

**Fig. 4 Kir2.1-mediated $V_m$ supports nutrient uptake for SAM generation to induce a metabolism-responsive inflammatory program. a** Schematic of SAM biosynthesis. **b, c** LC–MS analysis of SAM in mouse peritoneal macrophages ($n = 3$ and 4 respectively, mean ± SEM). **d** Schematic of derivation and contribution of glucose derived carbon atoms in SAM and ATP in mouse peritoneal macrophages. (red circles, [13C]-labeled carbon atoms; black, unlabeled carbon atoms). **e** Mass isotopomer distribution analysis of SAM and ATP derived from glucose in mouse peritoneal macrophages. ($n = 3$ and 5 respectively, mean ± SEM). **f** Mass isotopomer distribution analysis of SAM and ATP derived from glucose in mouse peritoneal macrophages. ($n = 2$ for control and $n = 5$ for others; mean ± SEM). **g** Schematic of derivation and contribution of serine-derived carbon atoms in SAM and ATP in mouse peritoneal macrophages. (red circles, [13C]-labeled carbon atoms; black, unlabeled carbon atoms). **h** Mass isotopomer distribution analysis of SAM derived from serine in mouse peritoneal macrophages. ($n = 3$ and 4 respectively, mean ± SEM). **i** IL-1β mRNA transcription analyzed by qPCR in mouse peritoneal macrophages. ($n = 3$, mean ± SD). **j** Volcano plot analysis of gene expression of SAM-dependent methyltransferase by RNA-seq in BMDMs. The expression levels of methyltransferase genes for H3K36me (highlighted)[39] among 183 annotated SAM-dependent methyltransferase genes[38] were analyzed. **k–m** ChIP-qPCR analysis of H3K36me3 enrichment in the *Il1b* gene in mouse peritoneal macrophages. ($n = 3$, mean ± SD). **n** ChIP-qPCR analysis of H3K36me3 enrichment in the *Il1a*, *Il18*, and *Cxcl10* genes in mouse peritoneal macrophages. ($n = 3$, mean ± SD). Two-tailed unpaired Student's *t*-test. The ChIP-qPCR data are representative of three independent experiments. Source data are provided as a Source Data file.

greatly suppressed IL-1β production in freshly isolated synovial fluid cells from gouty patients (Fig. 7i). These results indicate Kir2.1 as an important player driving inflammation triggered by both pathogenic and danger signals, and suggest a potential anti-inflammatory strategy by targeting Kir2.1.

## Discussion

Despite the fact that activated macrophages are not prone to rapid anabolic growth upon microbial infection or tissue damage, they still rapidly induce anabolic processes to support multiple pro-inflammatory functions as a hallmark of innate immunity. Nutrient import is the proximal step at which the increasing anabolic demands created for inflammation can be rapidly met[5]. Consistent with this idea, overexpression of GLUT1 is sufficient to increase the inflammatory character of macrophages[45]. However, the strategies adopted by activated macrophages to rapidly import nutrients such as glucose and amino acids from extracellular space are mysterious. In the present work, we propose an ionic-$V_m$ control of nutrient acquisition whereby inflammatory macrophages acquire a constant supply of molecular building blocks to support their inflammatory functions (Supplementary Fig. 11).

The apparent difference in the sensitivity to metabolic regulation of different inflammatory genes upon macrophage activation has been recognized for many years. For example, blocking glycolysis with 2-deoxyglucose (2-DG) greatly impairs LPS-induced IL-1β transcription, but not that of TNF[65]. Moreover, LPS-stimulated macrophages exhibit increased accumulation of the TCA cycle metabolites succinate, resulting in the induction of IL-1β but not TNF[2,65,66]. Previous studies have reported that a cluster of inflammatory genes are particularly responsive to glucose and amino acid metabolism, including *Il1b*, *Il1a*, *Il6*, *Il18*, *Il12a*, and *Cxcl10*, but not *Tnf*[4]. Crucially, the suppression of this cluster of metabolism-responsive genes was well recapitulated by Kir2.1 repression and $V_m$ depolarization. Ultimately, the cellular availability of SAM, whose generation and turnover are tightly controlled by the synergism of multiple nutrient inputs including glucose and amino acids, is a determinant of the $V_m$-dependent epigenetic regulation of this cluster of inflammatory factors, typified by IL-1β. Therefore, in addition to the multiple mechanisms that regulate the general signaling pathways essential for inflammation[67], we propose an additional electrochemical player that integrates extracellular nutrient inputs and intracellular epigenetic outputs in an inflammatory program that is highly dependent on nutrient acquisition.

Changes in environmental K$^+$ concentration or disturbances in the concentrations of ions have been reported in inflammation-related situations including sepsis and severe burns[68–71], thus it will be interesting to examine the membrane potential of macrophages and its impact on macrophage functions under these pathological conditions. Notably, PM depolarization has been found to stimulate K-Ras signaling for the proliferation of cancer cells[6], which provides another aspect of distinct adaptations in cancer cells and immune cells in response to the ionic disturbance within the tumor microenvironment[7,44], endowing them with different competing abilities to access the already limited nutrients in the tumor microenvironment.

## Methods

The research reported in this article complies with all relevant ethical regulations. The animal experimental protocols were approved by the Review Committee of Zhejiang University School of Medicine and were in compliance with institutional guidelines. To use these clinical materials for research purposes, the patients gave prior written informed consent as approved by the Institutional Research Ethics Committee of The Second Affiliated Hospital of Zhejiang University School of Medicine (approval no. 2018-064).

**Mice.** The background of all mice used was C57BL/6. C57BL/6 mice were purchased from the Model Animal Research Center of Nanjing University, Lyz2-cre mice were purchased from the Jackson Laboratory, and the *Kcnj2f/f* mice were kind gifts from Professor Mark T. Nelson of the University of Vermont. *Kcnj2f/f* mice were crossed with Lyz2-cre mice to obtain Lyz2-cre-*Kcnj2f/f* mice. Animals were housed in a specific pathogen-free facility maintained below 22 °C and 55% humidity under a 12-h light-dark cycle and free access to food and water in the University Laboratory Animal Center. The animal experimental protocols were approved by the Review Committee of Zhejiang University School of Medicine and were in compliance with institutional guidelines.

**Cells.** HEK293T cells were from the ATCC and cultured in Dulbecco's modified Eagle's medium (DMEM).

To obtain mouse peritoneal macrophages, on day 0, mice were injected intraperitoneally with 2.5 ml of 4% thioglycolate medium (Merck), and on days 3–5, peritoneal macrophages were obtained by flushing the peritoneal cavity with PBS or DMEM medium. Each mouse usually yielded ~$2 × 10^7$ cells, and the non-adherent cells were discarded after the macrophages adhered.

Bone marrow cells were flushed from tibias and femurs with cold DMEM and cultured in DMEM supplemented with 10% fetal bovine serum (FBS), 1% penicillin/streptomycin, and 10 ng/ml macrophage colony-stimulating factor (PeproTech) to generate BMDMs.

The iBMDMs were a kind gift from Prof. Shao (National Institute of Biological Sciences, China). iBMDMs were cultured in DMEM supplemented with 10% FBS and 1% penicillin/streptomycin. THP-1 cells were cultured in RPMI 1640 supplemented with 10% FBS and 1% penicillin/streptomycin.

Synovial fluid cells ($5 × 10^5$/well) were seeded in 12-well plates in RPMI 1640 supplemented with 10% FBS. They were stimulated with 100 ng/ml LPS and inhibitors as indicated for 12 h. Then sample supernatants were used for IL-1β measurements by ELISA.

**Patient-derived samples.** Synovial fluid (~4–5 ml) was obtained from two gouty patients (a 36-year-old man and a 56-year-old man) with serum uric acid levels >500 mmol/l and knee effusion. The patients were not involved in previous procedures or drug tests. To use these clinical materials for research purposes, the patients gave prior written informed consent as approved by the Institutional Research Ethics Committee of The Second Affiliated Hospital of Zhejiang University School of Medicine (approval no. 2018-064). Synovial fluid cells ($5 × 10^5$) were seeded in 12-well plates in RPMI 1640 supplemented with 10% FBS. They were stimulated with 100 ng/ml LPS and inhibitors as indicated for 12 h. Then sample supernatants were used for IL-1β detection using ELISA.

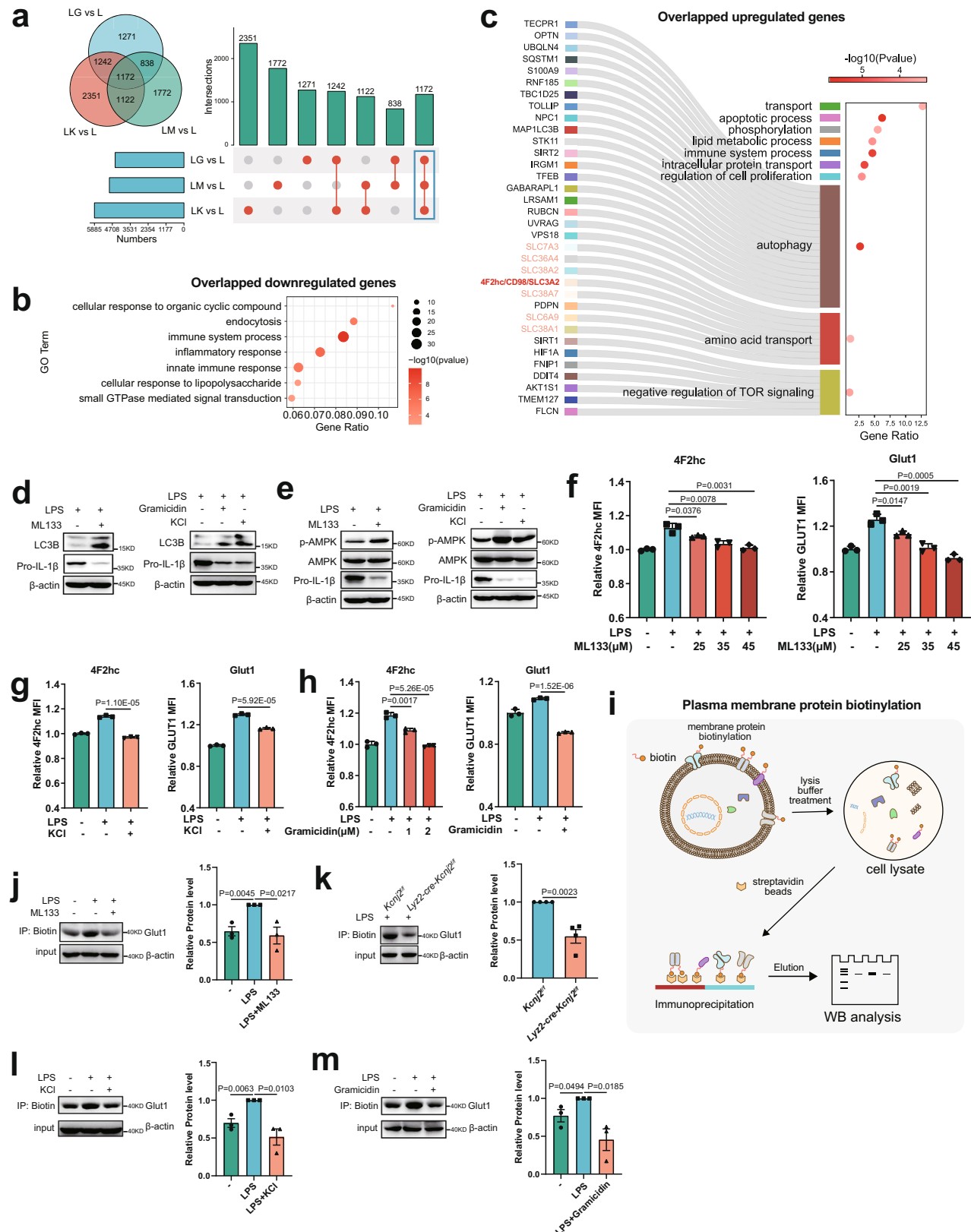

**Reagents**. LPS from *Escherichia coli* O111:B4, U-[$^{13}$C]-serine, serine, glycine, methionine, and S-(5′-Adenosyl)-L-methionine chloride dihydrochloride (SAM) were from Sigma; ML133 was from Selleck; gramicidin, PA-6, Tram-34, PAP-1 were from MedChemExpress; dialyzed FBS was from Biological Industries; and U-[$^{13}$C]-glucose was from Cambridge Isotope Laboratories; 2NBDG was from Cayman; Nystatin was from Sigma; cell permeable Dimethyl 2-Oxoglutarate (α-KG) was from 9 Ding Chemistry.

The following antibodies were used: pro-IL-1β (AF-401-NA, 1:1000, WB) was from R&D systems; anti-Kir2.1 (19965-1-AP, 1:1000, WB) was from Proteintech; anti-phospho-p65(#3033, 1:1000, WB), anti-p65(#8242, 1:1000, WB), anti-phospho-IκBα(#2859, 1:1000, WB), anti-IκBα(#4814, 1:1000, WB), anti-phospho-Erk(#4370, 1:1000, WB), anti-Erk(#4695, 1:1000, WB), anti-phospho-JNK(#9255, 1:1000, WB), anti-JNK(#9252, 1:1000, WB), anti-phospho-P38(#4511, 1:1000, WB), anti-P38(#8690, 1:1000, WB), anti-phospho-AMPK (#2535, 1:1000, WB),

**Fig. 5 Kir2.1-mediated $V_m$ drives nutrient uptake by retaining nutrient transporters on the macrophage cell surface. a–c** Overlap analysis of differentially expressed genes of mouse BMDMs treated with 500 ng/ml LPS in the presence or absence of ML133 (25 μM), elevated $[K^+]_e$ (50 mM), or gramicidin (1.25 μM) for 6 h (**a**), and pathway enrichment analysis of overlapped downregulated genes (**b**) and upregulated genes (**c**). LM: LPS + ML133; LG: LPS + Gramicidin; LK: LPS + KCl. **d, e** Western Blot analysis of autophagy (**d**) and AMPK activation (**e**) in mouse peritoneal macrophages treated with LPS in the presence or absence of ML133 (25 μM), elevated $[K^+]_e$ (50 mM), or gramicidin (1.25 μM) for 6 h. **f** Membrane GLUT1 and 4F2hc expression FACS assays of peritoneal macrophages treated with or without 500 ng/ml LPS in the presence or absence of different doses of ML133 for 2 h ($n = 3$, mean ± SD). **g, h** Membrane GLUT1 and 4F2hc expression FACS assays of peritoneal macrophages treated with or without 500 ng/ml LPS in the presence or absence of elevated $[K^+]_e$ (50 mM) or gramicidin (1 μM or 2 μM) for 2 h. ($n = 3$, mean ± SD). **i** Schematic of plasma membrane protein biotinylation and detection. **j–m** Western blots of membrane GLUT1 expression assays in peritoneal macrophages treated with or without 500 ng/ml LPS in the presence or absence of ML133 (25 μM), elevated $[K^+]_e$ (50 mM) or gramicidin (1.25 μM) for 2 h (left panel, representative result; right panel, statistics of bands intensity). Two-tailed unpaired Student's $t$-test. The FACS data are representative of three independent experiments. Gating strategies for FACS were shown in Supplementary Fig. 10. Source data are provided as a Source Data file.

anti-AMPK (#5831, 1:1000, WB), and anti-H3K36me3 (#4909, 1:200 CHIP) were all from Cell Signaling Technology; anti-LC3B (A19665, 1:1000, WB) was from ABclonal; anti-GLUT1 (ab115730,1:1000 for WB, 1:200 for flow cytometry and immunofluorescence), mouse monoclonal [2C11] to PIP2 (ab11039, 1:200, immunofluorescence), and goat polyclonal secondary antibody to mouse IgG - H&L (Alexa Fluor 647) (ab150115, 1:200, immunofluorescence) were from Abcam; PE anti-mouse 4F2hc (#128207, 1:200, flow cytometry), APC/Cyanine7 anti-mouse CD45 (103115, 1:100, Flow Cytometry), PE/Cyanine7 anti-mouse/human CD11b (101215, 1:100, Flow Cytometry), APC anti-mouse F4/80 Antibody (123115, 1:100, Flow Cytometry), and APC anti-mouse Ly-6C Antibody (128015, 1:100, Flow Cytometry) were from Biolegend; anti-4F2hc (15193-1-AP, 1:200, immunofluorescence) was from Proteintech; anti-β-actin (M1210-2, 1:3000, WB), anti-rabbit IgG-HRP (HA1001, 1:3000, WB), anti-mouse IgG-HRP (HA1006, 1:3000, WB), and anti-goat IgG-HRP (HA1005, 1:3000, WB) were from HuaBio; DyLight549 goat anti-rabbit IgG [H + L] (70-GAR5492) and DyLight488 goat anti-rabbit IgG [H + L] (70-GAR4882) were from MultiSciences; ChIP kit (#9005) was from Cell Signaling Technology; ELISA kit for IL-1β and IL-6 were from ThermoFisher; ELISA kit for IL-1α was from DAKEWE (1210112); and PI(4,5)P2 Mass ELISA kit was from Echelon (K-4500).

**Immunoblot analysis.** Cells were lysed in 2× SDS buffer (100 mM Tris-HCl, 4% SDS, 20% glycerol, 2% 2-mercaptoethanol, and 0.05% bromophenol blue) followed by boiling for 10 min. Then samples were separated by SDS-PAGE on 12% gels, after which the proteins were transferred to nitrocellulose membranes (#28637358, Pall). The membranes were blocked for 1 h in blocking buffer (5% skimmed milk and 0.1% Tween 20 in TBS), and then incubated with primary antibodies in 5% BSA overnight. The membranes were incubated with secondary antibodies in 0.1% Tween 20 in TBS at room temperature for 1 h. To detect proteins, we used ECL blotting reagents (Thermo Fisher).

**In vivo LPS challenge.** Mice were injected intraperitoneally with LPS (25 mg/kg body weight) alone or along with ML133 (30 mg/kg body weight). In the sepsis model, mice were sacrificed 4 h after LPS challenge, and the serum levels of IL-1β, IL-1α, and IL-6 were measured by ELISA according to the manufacturer's instructions. For mouse survival rate analysis, mice were injected intraperitoneally with LPS (20 mg/kg body weight) alone or along with ML133 (30 mg/kg body weight), then survival rates were analyzed.

**Bacterial infection.** For in vitro bacterial infection assay, mouse peritoneal macrophages were seeded in 12-well plates ($5 \times 10^5$/well) and infected with $5 \times 10^6$ E.coli or Salmonella SL1344 for 6 h in the presence or absence of ML133 (25 μM) followed by qPCR analysis of inflammatory gene transcription. For in vivo bacterial infection assays, 8-week-old mice were injected with $1 \times 10^7$ bacteria intraperitoneally in the presence or absence of ML133 (30 mg/kg); 6 h later, the mice were sacrificed and the serum levels of IL-1β was measured by ELISA according to the manufacturer's instructions.

**Quantitative PCR.** RNA was extracted using RNAiso Plus reagent (Takara). Complementary DNA was synthesized using HiScript II Reverse Transcriptase (Vazyme Biotech) according to the manufacturer's protocol. qPCR was performed using SYBR Green (Vazyme Biotech) on a CFX96 Touch Real Time PCR (BioRad). The PCR program was initial denaturation at 95 °C for 2 min, then cDNA amplification for 40 cycles at 95 °C for 30 s and 60 °C for 30 s. The samples were individually normalized to *Gapdh*. The primers are listed in Table 1.

**Measurement of extracellular acidification rate.** The Seahorse XF96 analyzer (Agilent Technologies) was used to measure the ECAR. BMDMs were seeded on XF96 plates at $8 \times 10^4$ cells/well one day prior to the XF assay. On the day of assay, the medium was replaced with assay medium composed of XF Base Medium without phenol red (Agilent Technologies, 103335-100) supplemented with 10 mM

glucose, 2 mM L-glutamine, and 1 mM sodium pyruvate, adjusted to pH 7.4 and incubated at 37 °C without $CO_2$ 45 min prior to XF assay. The assay protocol was as follows: baseline measurement with five cycles (mix 3 min, wait 0 min, measure 3 min); then LPS and/or ML133 was injected with a final concentration of 500 ng/ml or 25 μM, respectively, and measurements continued with 10–99 cycles (mix 3 min, wait 0 min, measure 3 min). Data shown are the mean ± SD and $n = 6$ technical replicates.

**[$^{13}$C]-glucose and [$^{13}$C]-serine tracing.** Mouse peritoneal macrophages ($5 \times 10^6$/dish) were seeded in 60-mm dishes in complete DMEM (supplemented with 10% FBS and 1% penicillin/streptomycin) to adhere. In metabolite tracing experiments, complete DMEM was replaced with glucose- or serine-deficient DMEM supplemented with 25 mM U-[$^{13}$C]-glucose (Cambridge Isotope Laboratories) or 0.4 mM U-[$^{13}$C]-serine (Sigma) (supplemented with 10% dialyzed FBS and 1% penicillin/streptomycin). Cells were treated with 500 ng/ml LPS in the presence or absence of compounds as indicated for 6 h. For metabolite extraction, cells were washed twice with PBS and once more with 0.9% NaCl. After completely aspirating the liquid, the plates were put on dry ice, 1 ml of 80% (v/v) methanol (pre-chilled to −80 °C) was added, and the plates were kept at −80 °C for 2 h. The plates were scraped on dry ice, the cell lysate/methanol mixture was transferred to a 2-ml tube on dry ice, then another 0.8 ml of 80% methanol was added to wash the plate and transfer the mixture to a tube. Each tube was centrifuged at 14,000 g for 20 min at 4 °C and the metabolite-containing supernatant was transferred to a new tube and lyophilized. Metabolites were analyzed using a TSQ Quantiva Ultra triple-quadrupole mass spectrometer coupled with an Ultimate 3000 UPLC system (Thermo Fisher, CA) equipped with a heated electrospray ionization probe. Chromatographic separation was done by gradient elution on a reversed-phase UPLC HSS T3 column ($2.1 \times 100$ mm, 1.7 μm; Waters). Mobile phase A consisted of 2 mM perfluoroheptanoic acid in 100% $H_2O$, and mobile phase B of 100% acetonitrile. A 10-min gradient at a flow rate of 300 μL/min was used as follows: 0–1.5 min at 2% B; 1.5–5 min, 2–98% B; 5–7 min, 98% B; 7–7.1 min, 2% B; 7.1–10 min 2% B. The column chamber was held at 45 °C and the sample tray at 10 °C. Data were acquired in Selected Reaction Monitoring in positive/negative switch ion mode and optimal transitions are reported in the table as indicated below. Both the precursor and fragment ions were separately collected at a resolution of 0.7 full width at half maximum. The source parameters were as follows: spray voltage, 3000 V; ion transfer tube temperature, 350 °C; vaporizer temperature, 300 °C; sheath gas flow rate, 35 Arb; auxiliary gas flow rate, 12 Arb. CID gas, 1.5 mTorr. Data analysis and quantitation were performed using Xcalibur 3.0.63 (Thermo Fisher, Carlsbad, CA). The liquid chromatography-mass spectrometry was done at the Metabolomics Facility of Technology Center for Protein Sciences, Tsinghua University, Beijing.

**Metabolite quantification and metabolomics.** Mouse peritoneal macrophages ($5 \times 10^6$/dish) were seeded in 60-mm dishes in complete DMEM (supplemented with 10% FBS and 1% penicillin/streptomycin) to adhere. Cells were treated with 500 ng/ml LPS in the presence or absence of compounds as indicated for 6 h. Then metabolites were extracted as above. For unbiased and targeted metabolomics, samples were analyzed at ChemDataSolution Information Technology (Dalian, China) and Applied Protein Technology (Shanghai, China).

For unbiased metabolomics, LC–MS/MS Analysis (HILIC/MS) were performed using a UHPLC (1290 Infinity LC, Agilent Technologies) coupled to a quadrupole time-of-flight (AB Sciex TripleTOF 6600). For HILIC separation, samples were analyzed using a 2.1 mm × 100 mm Acquity UPLC BEH 1.7 μm column (Waters, Ireland). In both ESI positive and negative modes, the mobile phase contained A = 25 mM ammonium acetate and 25 mM ammonium hydroxide in water and B = acetonitrile. The gradient was 85% B for 1 min and was linearly reduced to 65% in 11 min, and then reduced to 40% in 0.1 min and kept for 4 min, then increased to 85% in 0.1 min, with a 5-min re-equilibration period. The ESI source conditions were set as follows: Ion Source Gas1 (Gas1) as 60, Ion Source Gas2 (Gas2) as 60, curtain gas (CUR) as 30, source temperature: 600 °C, IonSpray Voltage Floating ±5500 V. In MS only acquisition, the instrument was set to acquire over the m/z range 60–1000 Da, and the accumulation time for the TOF

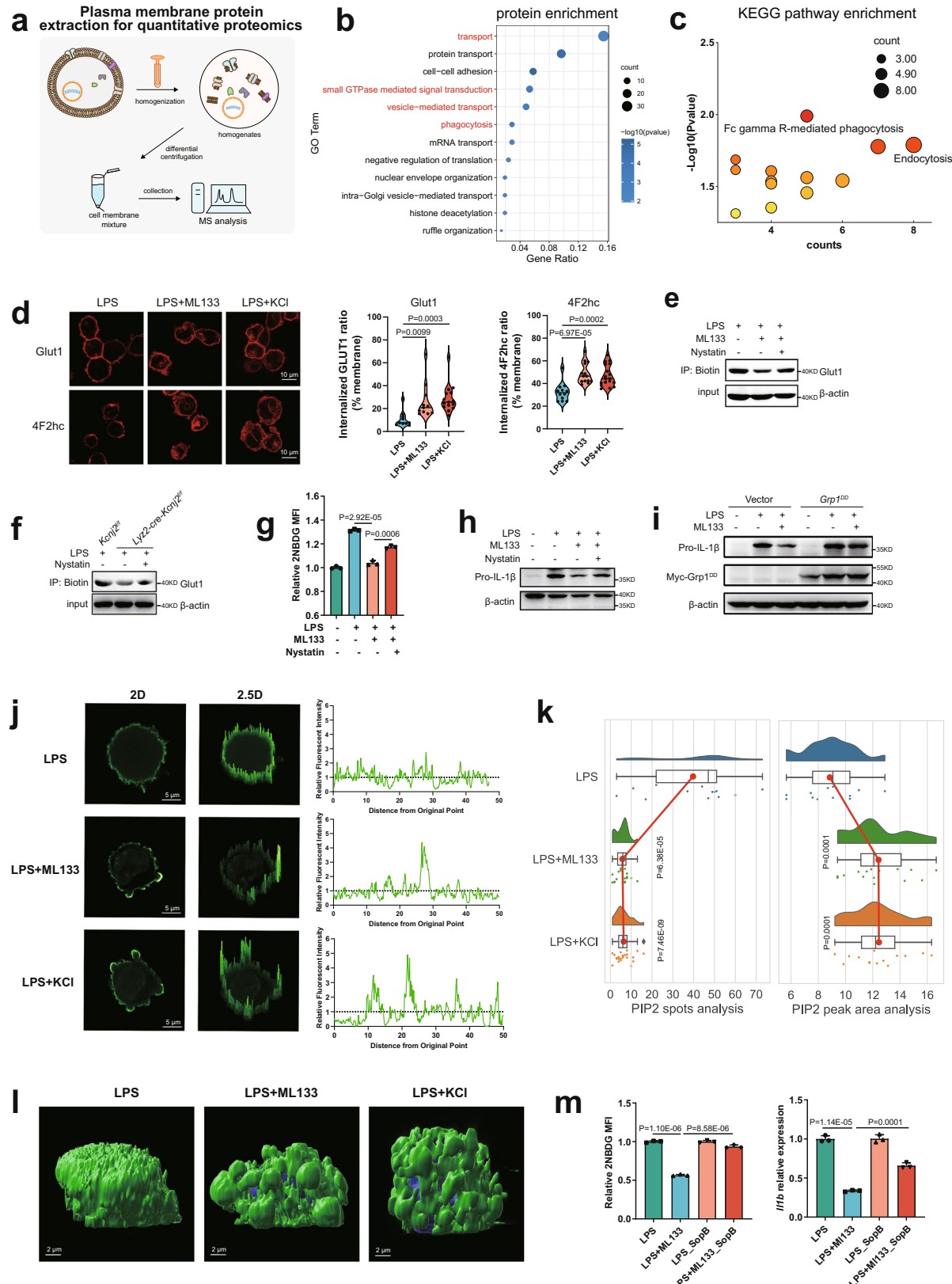

MS scan was set at 0.20 s/spectra. In auto MS/MS acquisition, the instrument was set to acquire over the m/z range 25–1000 Da, and the accumulation time for the product ion scan was set at 0.05 s/spectra. The product ion scan was acquired using information-dependent acquisition with high-sensitivity mode selected. The collision energy was fixed at 35 V with ±15 eV and the de-clustering potential was set at ±60 V.

A targeted metabolic analysis was performed using an LC–MS/MS system. The dried metabolites were dissolved in 100 μL of acetonitrile/ H$_2$O (1:1, v/v) and centrifuged at 16,200 × g rpm for 15 min. Electrospray ionization was conducted with an Agilent 1290 Infinity chromatography system and an AB Sciex QTRAP 5500 mass spectrometer. NH$_4$COOH (15 mM) and acetonitrile were used as mobile phases A and B, respectively. A binary solvent gradient was used as follows: A,

**Fig. 6 $V_m$ facilitates cell surface retention of nutrient transporters by configuring plasma membrane PIP$_2$ dynamics. a** Schematic of cell membrane protein extraction for quantitative proteomics analysis. **b, c** GO (**b**) and KEGG pathway (**c**) enrichment analysis of differentially expressed membrane proteins in peritoneal macrophages. **d** Confocal microscope analysis of membrane GLUT1 and 4F2hc internalization. (left, $n = 10, 11, 15$ respectively; right, $n = 12, 14, 18$ respectively). **e, f** Western blots of membrane GLUT1 expression in peritoneal macrophages treated with 500 ng/ml LPS in the presence or absence of 25 μM ML133 or 6 μg/ml nystatin for 2 h. **g** In vitro glucose uptake assays in mouse peritoneal macrophages. ($n = 3$, mean ± SD). **h** Western blots of pro-IL-1β expression in mouse peritoneal macrophages. **i** Western blots of pro-IL-1β expression in BMDMs overexpressing vector or the Grp1$^{DD}$ mutant treated with or without 500 ng/ml LPS in the presence or absence of 25 μM ML133 for 6 h. **j–l** Confocal microscope analysis of membrane PI(4, 5) P2 clustering in mouse iBMDMs. (left, the 2D and 2.5D images of PI(4, 5)P2 staining; right, membrane fluorescence of PI(4, 5)P2) (**j**). Statistics of membrane clustering of PI(4, 5)P2 showing better clustering of PIP2 (**k**) and 3D reconstruction of membrane PIP2 (**l**). ($n = 13, 19, 23$ respectively for the left panel; $n = 13, 15, 14$ respectively for the right panel; mean ± SD). **m** In vitro glucose uptake assays and IL-1β transcription analysis. ($n = 3$, mean ± SD). Two-tailed unpaired Student's $t$-test. The qPCR, western blot, and FACS data are representative of two or three independent experiments. Source data are provided as a Source Data file.

NH$_4$COOH; B, 0–18 min at 90–40% acetonitrile; 18–18.1 min at 40–90% acetonitrile; and 18.1–23 min at 90% acetonitrile. The LC–MS/MS was operated in the negative mode under the following conditions: source temperature, 450 °C; ion source gas 1, 45; ion source gas 2, 45; curtain gas, 30; and ion spray voltage floating, −4500 V.

**RNA-sequencing and differentially expressed genes analysis.** A total of 3 μg RNA per sample was used as input material for the RNA sample preparation. Sequencing libraries were generated using the NEBNext UltraTM RNA Library Prep Kit for Illumina (NEB, USA) following the manufacturer's recommendations, and index codes were added to attribute sequences to each sample. Briefly, mRNA was purified from total RNA using poly-T oligo-attached magnetic beads. Fragmentation was carried out using divalent cations under elevated temperature in NEB Next First Strand Synthesis Reaction Buffer (5×). First-strand cDNA was synthesized using a random hexamer primer and M-MuLV Reverse Transcriptase (RNase H$^-$). Second-strand cDNA synthesis was subsequently performed using DNA Polymerase I and RNase H. Remaining overhangs were converted into blunt ends via exonuclease/polymerase activity. After adenylation of the 3′ ends of DNA fragments, NEBNext Adaptor with hairpin loop structure was ligated to prepare for hybridization. In order to select cDNA fragments of preferentially 250–300 bp in length, the library fragments were purified with the AMPure XP system (Beckman Coulter, Beverly, USA). Then 3 μl USER Enzyme (NEB, USA) was used with size-selected, adaptor-ligated cDNA at 37 °C for 15 min followed by 5 min at 95 °C before PCR. Then PCR was performed with Phusion High-Fidelity DNA polymerase, Universal PCR primers, and Index (X) Primer. Finally, PCR products were purified (AMPure XP system) and library quality was assessed on the Agilent Bioanalyzer 2100 system.

Clustering of the index-coded samples was performed on a cBot Cluster Generation System using the TruSeq PE Cluster Kit v3-cBot-HS (Illumia) according to the manufacturer's instructions. After cluster generation, the library preparations were sequenced on an Illumina Hiseq platform and 125 bp/150 bp paired-end reads were generated.

FeatureCounts v1.5.0-p3 was used to count the read number mapped to each gene. And then the FPKM of each gene was calculated based on the length of the gene and the read count mapped to this gene. FPKM, the expected number of fragments per kilobase of transcript sequence per million base pairs sequenced, considers the effect of sequencing depth and gene length for the read count at the same time, and is currently the most commonly used method for estimating gene expression levels.

Differential expression analysis of two conditions/groups (two biological replicates per condition) was performed using the DESeq2 R package (1.16.1). DESeq2 provides statistical routines for determining the differential expression in digital gene expression data using a model based on the negative binomial distribution. The resulting $P$ values were adjusted using the Benjamin and Hochberg approach for controlling the false discovery rate. Genes with an adjusted $P$ value <0.05 found by DESeq2 were defined as differentially expressed.

Gene Ontology (GO) enrichment analysis of differentially expressed genes was implemented by the cluster Profiler R package, in which gene length bias was corrected. GO terms with a corrected $P$ value <0.05 were considered significantly enriched by differentially expressed genes.

GSEA (http://www.broadinstitute.org/gsea/index.jsp) of the expression data was used to assess enrichment of the KEGG as well as the SGOC gene-sets. The GSEA was performed using the OmicStudio tools at https://www.omicstudio.cn/tool.

**Electrophysiology.** Whole-cell current recordings were performed using a HEKA EPC10 amplifier controlled with PatchMaster software (HEKA) at room temperature with a 500-ms voltage ramp from −120 mV to +60 mV applied every 5 s. ML133 and KCl were added when the currents reached the steady state. Specifically, the extracellular solution (ECS) contained (in mM) 135 NaCl, 5 KCl, 2 CaCl$_2$, 1 MgCl$_2$, and 10 HEPES (adjusted to pH 7.4 with NaOH). For high K$^+$-induced depolarization, the external solution was changed to high K$^+$ ECS containing (in mM) 50 potassium L-aspartate, 90 NaCl, 2 CaCl$_2$, 1 MgCl$_2$, and 10 HEPES

(adjusted to pH 7.4 with NaOH). The pipette solution contained (in mM) 147 potassium L-aspartate, 2 MgCl$_2$, and 10 HEPES (adjusted to pH 7.3 with KOH). The membrane potential was held at −40 mV. Glass pipettes with a resistance of 3–5 MΩ were used. Data were acquired at 10 kHz and filtered offline during data analysis. The series resistance was <15 MΩ and monitored after experiments. The extracellular solution was changed using an RSC-200 system (Bio-Logic Science Instruments).

To continuously monitor changes in Vm, the whole-cell configuration was applied using current clamp ($I = 0$). we recorded a total of 3 min with or without ML133 and 5 s with high K$^+$ (50 mM). The membrane potential was the average $V_m$ within 30 s of initiation or the average $V_m$ of the last 30 s of a 3-min recording.

**Membrane potential assessment by FACS.** Mouse peritoneal macrophages were seeded into 12-well plates ($6 \times 10^5$/well). The cellular Vm-sensitive probe DiBAC$_4$(3) was mixed with the indicated compounds and the peritoneal macrophages were incubated for 1 h. Then the cells were washed three times with cold PBS and subjected to FACS analysis.

**Chromatin immunoprecipitation and ChIP-qPCR.** Cells were seeded at $1 \times 10^7$/ dish in 100-mm dishes in complete DMEM (supplemented with 10% FBS, and 1% penicillin/streptomycin) to adhere, then treated with 500 ng/ml LPS in the presence or absence of compounds as indicated for 6 h. Then experiments were performed according to the kit protocol (SimpleChIP Plus Enzymatic Chromatin IP Kit, CST#9005) and the final DNA samples were used for qPCR. DNA amplification consisted of 40 cycles at 95 °C for 30 s and 60 °C for 30 s. ChIP primers used in the study were shown in Table 2.

**Glucose uptake assay.** For in vitro glucose uptake assays, mouse peritoneal macrophages were seeded at $5 \times 10^6$/well in 12-well plates and treated as indicated. The cells were starved for 30 min at 37 °C in Krebs–Ringer bicarbonate solution (KRBH, in mM: 135 NaCl, 3.6 KCl, 0.5 NaH$_2$PO$_4$, 1.5 CaCl$_2$, 2 NaHCO$_3$, 10 HEPES, and 0.1% BSA) after washing three times with cold KRBH. Then the cells were incubated in KRBH supplemented with 2NBDG (60 μM) at 37 °C for 15 min. After washing three times with cold KRBH the cells were scraped for FACS analysis. For in vivo glucose uptake assays, 8-week-old mice were intraperitoneally injected with LPS (1 mg/kg); 30 min later 2NBDG (500 nmol/mouse) was injected, and after another 1 h, blood and peritoneal exudate cells were sequentially collected. Peripheral blood mononuclear cells were isolated from blood samples using Ficoll-Paque PLUS (GE Healthcare) according to the manufacturer's instructions. Peritoneal exudate cells (PECs) were flushed with 5 ml cold PBS. Macrophages (CD45$^+$ CD11b$^+$ F4/80$^+$ cells) in PECs and monocytes (CD45$^+$ CD11b$^+$ Ly6C$^{high}$ cells) in the blood were stained with the corresponding antibody for 30 min on ice. After washing in PBS, FACS analysis of 2NBDG MFI was carried out with a BD Fortessa Multicolor flow cytometer.

**Membrane protein biotinylation assay.** Mouse peritoneal macrophages were seeded at $4 \times 10^6$/dish in 60-mm dishes and treated as indicated. The cells were washed three times with ice-cold PBS to remove contaminating proteins. Then cell-surface proteins were biotinylated by incubating the cells with Sulfo-NHS-SS-Biotin solution at 1 mg/ml in PBS on ice. Two hours later, the cells were washed three times with ice-cold PBS to remove non-reacted biotinylation reagent. Alternatively, 25–50 mM Tris (pH 8.0) was used for the initial wash to quench non-reacted biotinylation reagent. After lysis and centrifugation at 13800 × g rpm for 10 min, the supernatant was transferred to a 1.5-ml microcentrifuge tube containing pre-washed streptavidin magnetic beads and incubated at 4 °C overnight with rotation. The streptavidin magnetic beads were washed and lysed in SDS-PAGE Reducing Sample Buffer and the samples were used for western blot.

**Plasma Membrane lipid extraction.** Experiment was performed as described[72]. Briefly, monolayer cells were washed twice at 4 °C with ice-cold PBS followed by

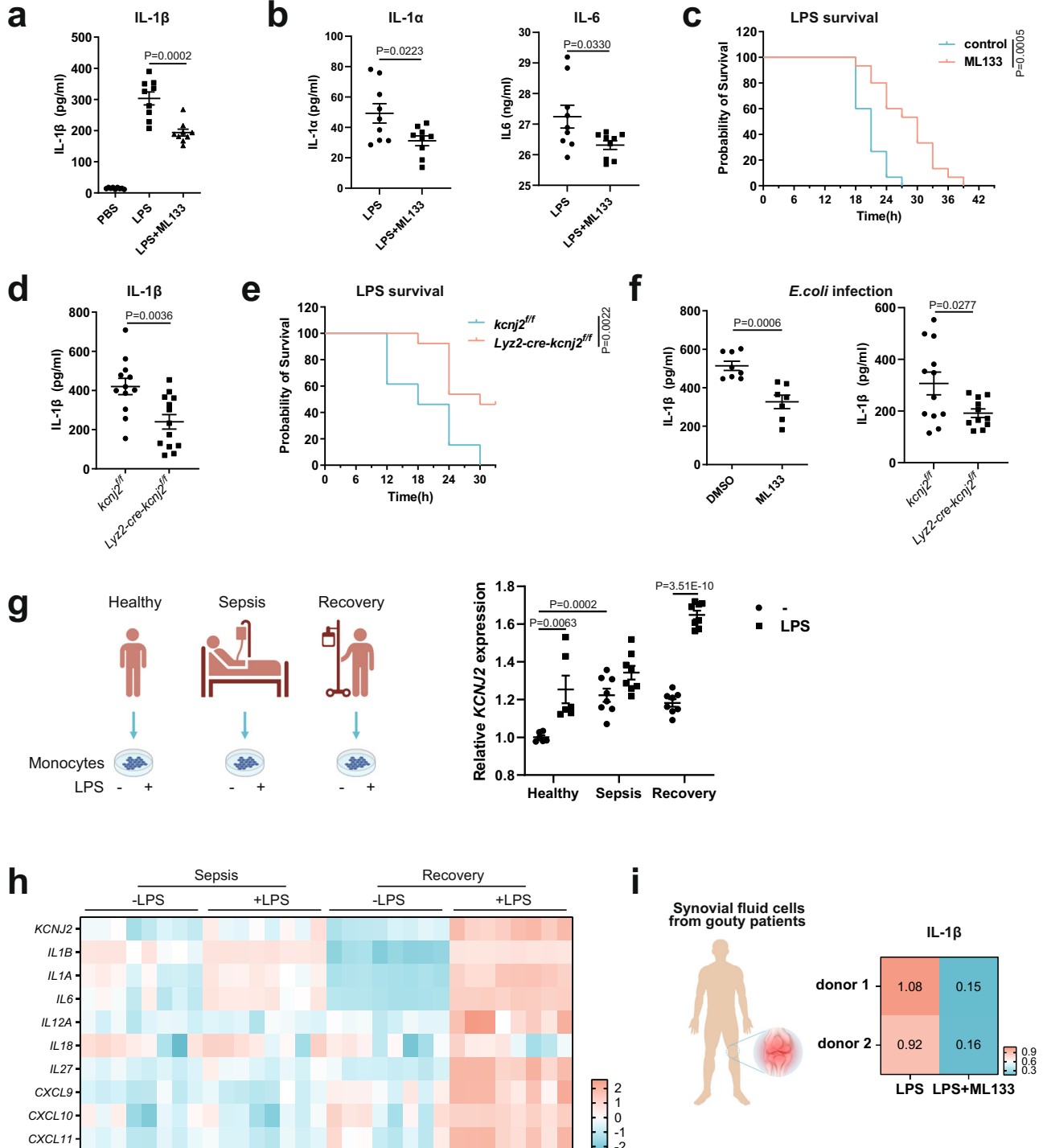

**Fig. 7 Kir2.1 repression alleviates inflammation triggered by pathogenic or danger signals in mouse models and human samples. a** IL-1β levels in serum from septic mice measured by ELISA. (*n* = 9; mean ± SEM) (**b**) IL-1α and IL-6 levels in serum from septic mice measured by ELISA. (*n* = 9, mean ± SEM) (**c**) Survival rates of sepsis model mice (*n* = 15, log-rank test [Mantel–Cox]). **d** IL-1β levels in serum from septic mice measured by ELISA (*Kcnj2f/f*, *n* = 12; *Lyz2-cre-Kcnj2f/f*, *n* = 13; mean ± SEM). **e** Survival rates of sepsis model mice (*n* = 13; log-rank test [Mantel–Cox]). **f** IL-1β levels in serum from septic mice measured by ELISA (DMSO, *n* = 8; ML133, *n* = 7; *Kcnj2f/f*, *n* = 12, *Lyz2-cre-Kcnj2f/f*, *n* = 11; mean ± SEM). **g** The expression of *KCNJ2* in monocytes from healthy donors, sepsis patients, and those who recovered from sepsis with or without LPS[63]. Data sourced from the GEO database (GSE46955). (*n* = 6, 8, 8 respectively; mean ± SEM). **h** Heatmap showing the expression of Kir2.1 and inflammatory genes in monocytes from septic patients and those who recovered from sepsis in response to LPS[63]. Data sourced from the GEO database (GSE46955). (*n* = 8, 8 respectively; mean ± SEM). **i** Relative IL-1β levels in the supernatant of cultured synovial fluid cells from two gouty patients analyzed by ELISA. Two-tailed unpaired Student's *t*-test. Source data are provided as a Source Data files.

**Table 1 Primers for q-PCR.**

| Gene | Forward 5′ to 3′ | Reverse 5′ to 3′ |
|---|---|---|
| Gapdh | AGGTCGGTGTGAACGGATTTG | TGTAGACCATGTAGTTGAGGTCA |
| Il1b | ATCAACCAACAAGTGATATTCTCCAT | GGGTGTGCCGTCTTTCATTAC |
| Tnf | CCTGTAGCCCACGTCGTAG | GGGAGTAGACAAGGTACAACCC |
| Il1a | GGAGAAGACCAGCCCGTGTTGCT | CCGTGCCAGGTGCACCCGACTT |
| Il6 | TAGTCCTTCCTACCCCAATTTCC | TTGGTCCTTAGCCACTCCTTC |
| Il12a | CTCCCTTGGATCTGAGCTGG | GCTGTTGGAACGCTGACCATA |
| IL18 | GACTCTTGCGTCAACTTCAAGG | CAGGCTGTCTTTTGTCAACGA |
| Cxcl10 | CCACGTGTTGAGATCATTGCC | GAGGCTCTCTGCTGTCCATC |

**Table 2 Primers for ChIP-qPCR.**

| Primer | Forward 5′ to 3′ | Reverse 5′ to 3′ | Source |
|---|---|---|---|
| Il-1b -ChIP-1126 | AGGGACTCCTACAGATGCAATGGT | TGCTCTGGTTGCTCTCTGTTGACT | Adamik et al., 2013 |
| Il-1b -ChIP-3627 | AAATCCAATGTTCTTGCCCAGCCC | TGCAAGCACTGTGAAGTGAAGCAG | Adamik et al., 2013 |
| Tnf -ChIP | AAAGAAGCCGTGGGTTGGACAGAT | AGAACTGATGAGAGGGAGGCCATT | Adamik et al., 2013 |
| Il1a-ChIP | AGGCTGATCAAGTCAACGGC | CTGATACCGGCCAGAAGGAC | ChIPprimersDB |
| Il18-ChIP | AGAGCCTTTGGGCTTTCTCC | GGTTTGGGACTTCGCTGGTA | ChIPprimersDB |
| Cxcl10-ChIP | GGGAGAGGGAAATTCCA | TTTCCCTCCCTGAGTCC | ChIPprimersDB |

incubation with PBS containing 1 mM MgCl$_2$ and 0.1 mM CaCl$_2$ for 15 min. Then Buffer B containing 0.5 mg/mL sulfo-NHS-SS-biotin was added to each monolayer, and the dishes were shaken for 1 h at 4 °C. The biotinylation reaction was stopped by removing buffer B followed by incubation with buffer C for 15 min at 4 °C. The cells were then scraped in buffer A and centrifuged at 1000 × g for 10 min at 4 °C. The resulting pellet was re-suspended in buffer D supplemented with a protease inhibitor mixture. The cells were lysed by passage 13 times through a ball-bearing homogenizer with a 10-μm clearance. The homogenate was centrifuged at 3000 × g for 10 min at 4 °C, after which the supernatant was incubated for 60 min at room temperature with streptavidin magnetic beads that were pre-equilibrated with buffer E. The beads were isolated by adherence to the tube wall in a magnetic field, after which they were washed for 10 min with buffer D. Then seven 10-min washes with buffer F supplemented with protease inhibitor mixture were followed by one quick wash with PBS. The beads were then de-magnetized, and lipids were extracted with hexane/isopropanol (vol/vol, 3:2) from washed plasma membranes attached to streptavidin magnetic beads. The organic solvent containing the extracted lipids was evaporated under a gentle stream of nitrogen and used for the measurement of lipids. The buffers used are listed below (in mM unless otherwise indicated):

Buffer A: 50 Tris·HCl (pH 7.5), 150 NaCl; Buffer B: 25 Hepes-KOH (pH 7.4), 150 NaCl, 2 CaCl$_2$; Buffer C: 50 Tris·HCl (pH 7.5), 150 NaCl, 100 glycine; Buffer D: 50 Tris·HCl (pH 7.5), 10 KCl, 1.5 MgCl$_2$, 1 sodium EDTA, 1 sodium EGTA, 250 sucrose; Buffer E: 25 Hepes-KOH (pH7.4), 150 NaCl, 0.2% (wt/vol) BSA; Buffer F: 50 Tris·HCl (pH 7.5), 5 μM biotin.

**Plasma membrane proteome analysis.** Plasma membrane protein extracted as described[73]. Briefly, cells were washed with ice-cold PBS and collected after centrifugation at 1200 × g for 5 min. The supernatant was discarded, 450 μl Buffer A (in mM: 250 sucrose, 20 HEPES, 10 KCl, 1.5 MgCl$_2$, 1 EDTA, 1 EGTA, protease cocktail) was added to re-suspend the cell pellets, and the cells were lysed 20 times with a 29 G needle. Buffer A was added to a volume of 1 ml and centrifuged at 700 × g for 5 min. The supernatant was transferred to a new tube and centrifuged at 10000 × g for 10 min. The supernatant was transferred to a new tube and centrifuged at 100,000 × g for 1 h. The supernatant was discarded and the membrane proteins were subjected to MS analysis. The proteome analysis was done in PTM BIO (Hangzhou, China).

**Immunofluorescence Staining and Confocal Microscopy.** For immunofluorescence staining of GLUT1 and 4F2hc, PMA-differentiated THP-1 cells were treated with 500 ng/ml LPS in the presence or absence of elevated [K$^+$]e (50 mM) and ML133 (25 μM) for 2 h. After washing twice with PBS, cells were fixed in 4% paraformaldehyde in PBS for 15 min and then washed three times in PBS. After permeabilization with saponin and blocking with 5% bovine serum albumin in PBS, cells were incubated with primary antibodies overnight at 4 °C. After washing with PBS, cells were incubated with secondary antibodies in PBS for 30 min and rinsed in PBS. Analyses were carried out using a confocal microscope Zeiss LSM 800. The internalized ratio of GLUT1 and 4F2hc were analyzed using ImageJ.

Staining of PI(4,5)P2 was performed as described[74]. Briefly, iBMDMs were treated with 500 ng/ml LPS in the presence or absence of elevated [K$^+$]e (50 mM) and ML133 (25 μM) for 2 h. After washing twice with PBS, Cells were rapidly fixed by 4% FA and 0.2% GA (glutaraldehyde) in PBS. Fixation was allowed to proceed for 15 min at room temperature (20–24 °C). Then cells were rinsed three times with PBS containing 50 mM NH$_4$Cl.Slides were then placed on a metal plate in a deep ice bath and chilled for at least 2 min. All subsequent steps were performed on ice, with all solutions pre-chilled. After that, cells were blocked and permeabilized for 45 min with a solution of buffer A (20 mM Pipes, pH 6.8, 137 mM NaCl, 2.7 mM KCl) containing 5% (v/v) NGS (normal goat serum), 50 mM NH$_4$Cl and 0.5% saponin. Primary antibodies were applied in buffer A with 5% NGS and 0.1% saponin for1 h. After two washes in buffer A, a 45 min incubation with secondary antibody in buffer A with 5% NGS and 0.1% saponin was performed. Slides were rinsed four times with buffer A, and cells were post-fixed in 2% FA in PBS for 10 min on ice, before warming to room temperature for an additional 5 min. FA was removed by three rinses in PBS containing 50 mM NH$_4$Cl, followed by one rinse in distilled water. Cells were then stained with DAPI (4,6-diamidino-2-phenylindole) and covered with glass cover slips, and sealed with nail varnish. Analyses were carried out using a confocal microscope Zeiss LSM 800. Membrane PIP2 fluorescence intensity were analyzed using ZEN2 blue edition 2.0.0.0. For membrane PIP2 fluorescence clustering analysis, Imaris was used to quantity the number of PIP2 fluorescence spots. The membrane PIP2 spots and PIP2 peak area of fluorescence were to show if PIP2 were continuously and uniformly distributed. The more PIP2 spots indicates better continuity of PIP2. The large peak area indicates the better clustering of PIP2. 3D reconstruction of membrane PIP2 were performed using Imaris.

**Flow cytometry.** Mouse peritoneal macrophages (6 × 10$^5$ cells) were seeded and stimulated under the indicated conditions. Then the cells were collected with a cell scraper, followed by antibody staining for 30 min at 4 °C. After washing with PBS to exclude non-specific staining, the surface expression of CD98 or GLUT1 was detected by flow cytometry (ACEA NovoCyte).

**Statistics and reproducibility.** For RNA-seq, pathway enrichment analysis of differentially expressed genes (DEGs) was performed in Metascape (a gene annotation and analysis resource) and DAVID Bioinformatics Resources. Metabolomic analysis was performed in MetaboAnalyst5.0, based on the KEGG and SMPDB databases. Data were processed in RStudio or bioinformatics tools (hiplot.com.cn and www.bioinformatics.com.cn). The results for q-PCR are expressed as the mean ± SD. The mouse sepsis model and the LC/MS experiments are expressed as the mean ± SEM and were analyzed using two-tailed Student's t-test for two groups. The q-PCR results are representative of at least three independent experiments. For mouse survival rate analysis, GraphPad Prism7 was used to plot Kaplan–Meier survival curves and to compare survival using log-rank tests.

**Reporting summary.** Further information on research design is available in the Nature Research Reporting Summary linked to this article.

## Data availability

The RNA-seq data generated in this paper have been deposited in the Gene Expression Omnibus (GEO) with primary accession number GEO: GSE146158 and GSE183067. The

mass spectrometry proteomics data have been deposited to the ProteomeXchange Consortium via the PRIDE partner repository with the dataset identifier PXD033322. The previously generated blood monocytes sequencing data reused in this study are available from GEO with primary accession number GSE46955. Source data are provided with this paper.

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

## Acknowledgements

We thank Prof. Mark T. Nelson for kindly providing the *Kcnj2$^{f/f}$* mice and Dr. IC Bruce for reading the manuscript. We thank Dr. Yudong Zhou for helpful discussion.

This work was supported by the National Natural Science Foundation of China (81930042, 81730047, 82025017 to D.W.; 81630091 and 31670840 to Y.W.; 32000630 to W.Y.; 32100692 to Z.C.; 32100721 to Y.G.). The work was also supported by China National Postdoctoral Program for Innovative Talents (BX20200296) and China Post-doctoral Science Foundation (2021M692780). This work was also supported by the Key Laboratory of Immunity and Inflammatory Diseases of Zhejiang Province and the Fundamental Research Funds for the Central Universities (2021-KYY-A07043-0009). We are grateful for the support of the Metabolomics Facility at the Technology Center for Protein Sciences, Tsinghua University. Thanks for the technical support by the Core Facilities, Zhejiang University School of Medicine. This work was also supported by the Key Laboratory of Immunity and Inflammatory Diseases of Zhejiang Province.

## Author contributions

W.Y., Z.W., X.Y., Y.Z., Z.X., Z.C., K.Z., S.C., T.X., D.J., X.G., M.L., J.Z., H.F., Y.G., and D.Y. performed the experiments; X.Y., X.Z., Y.W., W.Y., and D.W. designed the research; W.Y., Z.W., and D.W. wrote the manuscript; and D.W. supervised the project.

## Competing interests

The authors declare no competing interests.
