## [Peer Review File · Nature Communications]

Kir2.1-mediated membrane potential is a critical determinant of nutrient acquisition feeding inflammationEditorial Note: This manuscript has been previously reviewed at another journal that is not operating a transparent peer review scheme. This document only contains reviewer comments and rebuttal letters for versions considered at *Nature Communications*.

Reviewer #5 [macrophage, ion channel biology] (Remarks to the Author):

1. This work demonstrated an important role of the background potassium channel Kir2.1 in macrophages in the ionic control of metabolic-epigenetic reprogramming which links inflammation to nutrient acquisition. Authors performed a series of experiments to demonstrate the role of Kir2.1 in LPS-induced macrophage reprogramming and LPS-induced inflammation. However, there are critical questions that were not addressed.

We appreciate the reviewer's following comments and questions. After one year revision with additional experiments, this study has been largely strengthened and we believe it would be more convincing and rigorous to you.

2. Since Kir2.1 is continuously active or open even in unstimulated condition, (resting state), why was a membrane potential (MP) change (depolarization) not recorded in the baseline non-LPS treated cells but when the cells were treated with ML133 or in Kir2.1 deficient macrophages (Lyz2-cre-Kcnj2f/f)? Does this mean there is a compensatory mechanism for the Kir2.1 loss? If so, why does this apparent compensatory mechanism not function in LPS stimulated state in macrophage from Lyz2-creKcnj2f/f mice? Does LPS challenge annul the compensate mechanism?

3. The authors show no direct effect of LPS on both Kir2.1 current and the membrane potential, even on Kir2.1 expression (Figure S3B, treated with LPS for as long as 20 min), why does the Kir2.1 inhibitor, ML133, only polarize MP in cells pretreated with LPS? This is a key question. The authors need to explore this and define the mechanism. It is important to define a clear link between LPS challenge and the role of Kir2.1 in regulating membrane potential and macrophage reprogramming.

These two points raised a key issue we didn't well addressed in our previous version of manuscript, and it also confused us given the well-known function of Kir2.1 in regulating resting Vm (PMID:20086079, 4838798, 6323703). After a series of careful additional experiments examining the membrane potential of primary macrophages, we have now clearly concluded that blockade or absence of Kir2.1 depolarized both resting and LPS-stimulated macrophages. Using both patch clamp and membrane potential sensitive fluorochrome DiBAC4(3) (PMID: 27626381), macrophage Vm was depolarized when treated with ML133 alone for 1h (panel E, patch clamp) or 1h and 6h (panel A, Vm fluorochrome). Consistently, macrophages from Lyz2-cre-Kcnj2^{ff} mice showed depolarized membrane potential compared to controls (panel B, F, G). In LPS-stimulated macrophages, ML133 treatment still depolarized the Vm when stimulated

by LPS for 1h, 3h, and 6h (panel C). These results indicate that Kir2.1 plays a critical role in regulating V_m of both resting and inflammatory macrophages. In Figures 2E, 2F, and S2F (also indicated below in panel D and G), LPS stimulation had limited effect on Kir2.1 current and macrophage V_m , but ML133 treatment or Kir2.1 depletion greatly abolished Kir2.1 current and depolarized V_m independent of LPS treatment (panel D and F). Given the non-excitable characteristic of macrophages, we conclude that Kir2.1 acts as a basal and key regulator that maintains macrophage V_m . Although acute LPS stimulation has relatively small effect on Kir2.1 expression, current, and V_m , the metabolic demand of macrophage activation supplied by extracellular nutrients uptake is largely increased. Based on our following data, acute LPS stimulation increased glucose and serine uptake as well as their metabolic fluxes into the generation of SAM. To meet this LPS-stimulated anabolic metabolism, Kir2.1-mediated V_m plays a critical role in driving nutrients uptake by facilitating cell-surface retention of key nutrient transporters including 4F2hc/CD98 and GLUT1, which depends on the V_m -mediated configuration of plasma membrane phospholipid dynamics. We also need to emphasize that this study is focused on the regulation of immunometabolism by V_m in inflammatory macrophage which means the LPS stimulation is acute but not persistent or chronic, for example >12h. Because macrophage function and immunometabolism can be divided into two distinct stages during LPS stimulation (early stage: inflammatory vs. late stage: tolerate) which have almost opposite metabolic remodeling (glycolysis vs. oxPHOS) (PMID: 32259027, 28396078). The function and mechanisms of Kir2.1-mediated V_m would be changed in the latter stage but which is obviously beyond our scope.

4. The key mechanism claimed is that the cell plasma membrane expression of nutrient transporters, including GLUT1 and CD98, is regulated by Kir2.1-controlled membrane potential. The question how the membrane potential changes regulate these transporter surface expression needs to be explored.

We value and appreciate this important question, and performed further investigation to address this mechanistic weakness. Given the analogous phenotypes in the metabolic and epigenetic reprogramming and a subsequent metabolism-responsive inflammatory program under different Vm depolarization conditions (ML133, gramicidin, or elevated [K⁺]_e treatment), we reasoned that there would be common changes in the gene transcripts representing a specific mechanism underlying the Vm control of transporter surface expression and immunometabolism. To explore this, we performed combined analysis with the RNA-seq data from LPS-stimulated

macrophages in the presence of ML133, gramicidin, or elevated $[K^+]_e$ (LM, LG and LK group respectively). This analysis revealed 1172 differentially-expressed genes shared by all three treatments.

Further analyses demonstrated that the common upregulated genes upon Vm depolarization were significantly enriched in key pathways of metabolic adaptations in response to nutrient starvation such as 'autophagy', 'amino acid transport', and 'negative regulation of mTOR signaling', which is consistent with the impaired surface transporter expression and the subsequent nutrient starvation. In contrast, besides the pathways of 'inflammatory response' which are in line with the anti-inflammatory phenotype upon Vm depolarization, the common downregulated genes were also enriched in 'endocytosis', providing a possible mechanistic link between Vm depolarization and the loss of surface nutrient transporters.

To gain a deeper understanding of the impact of Vm depolarization on the molecular events occurring around the cell surface, we extracted the plasma membrane proteins of LPS-stimulated macrophages for further quantitative proteomics analysis (panel A). 215 differentially-expressed proteins were shared between the treatments with ML133 and elevated $[K^+]_e$ (panel B), and the pathways related to membrane trafficking including 'vesicle-mediated transport', 'endocytosis' and 'phagocytosis' were significantly enriched (panel C, D). These data are consistent with the view that nutrient uptake is tightly controlled by the dynamic membrane trafficking of such

transporters (PMID: 23402769).

A Plasma membrane protein extraction for quantitative proteomics

B

C

D

Consistent with these bioinformatics analyses, after ML133 or elevated [K⁺]_e treatment, LPS-stimulated macrophages showed significantly enhanced internalization of both GLUT1 and 4F2hc/CD98, providing the mechanistic explanation of the downregulation of these transporters surface expression.

Essentially, a depolarized V_m leads to changes in the electrochemical properties of the plasma membrane, which is composed mainly of charged phospholipids. V_m depolarization has recently been recognized to affect plasma membrane phospholipid dynamics, particularly the distribution of phosphatidylinositol 4,5-bisphosphate (PIP₂) (PMID: 26293964), whose enrichment in plasma membrane microdomains plays a central role in membrane trafficking and endocytosis by triggering plasma membrane invaginations and recruiting endocytic factors (PMID: 21613550, 26232664, 17035995,

19317650, 25264171). Although the amount of plasma membrane PIP2 was unchanged (panel A, B), we found that ML133 treatment or elevated $[K^+]_e$ led to significantly enhanced PIP2 clustering on the PM of LPS-stimulated macrophages by immunofluorescent staining with anti-PIP2 antibody and three-dimensional imaging (panel C, D, E). More importantly, the ML133-induced decreases in glucose uptake and IL-1 β production were significantly restored by the expression of lipid phosphatase SopB (panel F), an effective means of inhibiting endocytic trafficking by depleting the membrane level of PIP2 (PMID: 18411250, 20542249, 12360287). Collectively, these data suggest a critical regulation by which Kir2.1-mediated Vm modulates membrane phospholipid dynamics for the surface retention of nutrient transporters, thus ensuring nutrient acquisition to fuel inflammation.

All these additional results have been supplemented in the revised manuscript and we believe that now the manuscript is more complete particularly in term of mechanism study.

5. In the introduction, the authors should introduce; Kir2.1; it belongs to the background/leaking K⁺ channel family which is continuously active in cell resting state.

We appreciate this helpful suggestion and have added this description in the introduction.

6. Figure 1 including Figure S1. No ML133 time course control should be displayed to exclude IRK1 current decay with time. Also, 50 mM and 140 mM extracellular K⁺ against 140 mM intracellular K⁺ should be added to show the shift of reversal potential. Under these conditions, the MP is polarized by the increased extracellular K⁺. The similar condition should also be tested in the functional study as author did in Figure 6, such as LPS-induced signaling including GLUT1 and CD98 surface expression, generation of SAM and macrophage reprogramming index like IL-1 β .

As suggested, we performed patch clamp experiment to check whether Kir2.1 current decayed and how is the current decay rate. As shown below, current of Kir2.1 kept stable in 2 minutes (panel A), but ML133 treatment almost inhibited all Kir2.1 current in 2 minutes (panel B), which demonstrated that the inhibition effect was indeed caused by ML133 treatment rather than a decay of Kir2.1 currents over time.

To show the shift of reversal potential, we used 50 mM [K⁺]_e, a commonly used concentration to depolarize the membrane. As shown below, the membrane potential was depolarized when extracellular K⁺ changed from 5 mM to 50 mM (panel A). And the reversal potential also shifted to a more positive state (panel B, C).

As suggested, we also performed multiple functional studies under the condition of increased extracellular K⁺ treatment (50 mM), and obtained similar results as those by Kir2.1 blockade or genetic depletion. As shown below, these functional readouts included: 1) membrane PIP2 clustering, 2) Glut1 and 4F2hc/CD98 internalization, 3) the expression level of membrane surface Glut1 and 4F2hc/CD98, 4) cell metabolic adaptation, 5) metabolites isotopic tracer, 6) macrophage epigenetic reprogramming, and 7) metabolic-responsive genes transcription. These results are also supplemented and described in the revised manuscript.

1. membrane PIP2 clustering

2. Glut1 and 4F2hc/CD98 internalization

3. surface expression of membrane Glut1 and 4F2hc/CD98

4. cell metabolic adaptation

5. metabolites isotopic tracer

6. macrophage epigenetic reprogramming

7. metabolic-responsive genes transcription

7. Figure 2. Why were the cells not pretreated with LPS for 6h in Figure 1 (especially for membrane potential measurement), but here cells were pretreated with LPS for 6 h? LPS treatment for 1h does not affect Kir2.1 expression, but 6h inhibits Kir2.1 expression (based on the figure in rebuttal letter). If LPS pretreatment for a longer time (such as 12 h) could totally inhibit Kir2.1 expression as author displayed in the rebuttal letter, why did ML133 (the claimed Kir2.1 specific inhibitor) treatment still increase survival rate shown in Figure 2G? If LPS exerts a similar inhibitory effect on Kir2.1 expression, what causes the difference in the survival rate in Figure 2I between Kcnj2f/f and Lyz2-cre-Kcnj2f/f mice treated with LPS? A control with only ML133 treated group should be shown in Figure 2G.

We appreciate the reviewer for raising the issue of time points of detecting different readouts. In the working model based on our results, there is indeed a temporal sequence among the readouts we examined. The membrane potential depolarization was the earliest upstream event upon Kir2.1 blockade, which triggered the membrane PIP2 clustering and downstream membrane nutrients transporters internalization. Sequentially, this resulted in cellular nutrients starvation which further induced disturbed metabolic and epigenetic reprogramming including SAM production and transcriptional regulation of H3K36me3 on the metabolic-responsive inflammatory genes such as IL-1b. To make it better to understand the working model of this study, we marked the order of these biological events in the working model as indicated below (panel A). Therefore, we chose 1 h of LPS and ML133 treatment for membrane potential detection, 2h for nutrient transporters internalization detection, 6h for the

downstream metabolic status and gene transcription detection. However, as indicated in our response to Question 1 and data below, we found the effect of Vm depolarization by Kir2.1 blockade or depletion remained similar throughout the early stage of LPS stimulation (1h, 2h, 3h, 4h, 5h, and 6h) (panel B). Consistently, the inhibitory effect of IL-1 β production by ML133 was also evident throughout the early stage of LPS stimulation (panel C). We thus displayed certain *in vitro* readouts of representative time points.

As mentioned above, long time LPS treatment (>12h) would induce the transition of both metabolic and functional states of macrophages into another distinct stage. Although long time LPS treatment may affect Kir2.1 expression, the inhibition of inflammatory cytokines like IL-1 β at the early stage of LPS challenge has a profound influence on mouse survival. In addition, the *in vivo* readout such as the survival rate in a LPS-stimulated sepsis model is a cumulative process of inflammatory cytokines release. Moreover, the death of mice is largely dependent on a secondary attack of inflammatory cytokines targeting important organs, which follows their initial release to the circulation. Therefore, the time points of mice survival rate are not comparable to those of *in vitro* experiments.

As suggested, a control with only ML133 treated group was also done for this experiment and found no influence on mouse survival without LPS challenge.

8. Figure 3: Why was 30 min used for LPS pretreatment in Figure 3 EFG, instead of 6 h as in Figure 3 BC?

Please refer to our response to Question 7. Figure 3 EFG were glucose uptake experiments, Figure 3 BC were metabolomic experiments. The reason why different time points were used were explained as above. Consistently, we also performed the glucose uptake experiments with LPS and ML133 stimulation for 3 h and obtained similar results as indicated below.

REVIEWERS' COMMENTS

Reviewer #5 (Remarks to the Author):

The paper explores the effects of changing membrane potential on inflammatory function of macrophages. Studies are made using LPS stimulated peritoneal macrophages or bone marrow macrophages in which membrane potential was measured by patch clamping. Increasing extracellular potassium ion concentration or applying gramicidin that elevated extracellular potassium resulted in membrane depolarization, promoted nutrient uptake which in turn activated glycolytic, pentose phosphate pathway, citric cycle, purine synthesis, phospholipid biosynthesis, amino acid metabolism, and biosynthesis. These results support the notion that PM depolarization promotes nutrient acquisition and subsequent anabolic metabolism and makes macrophages better host defense cells.

Of the 54 annotated potassium channel genes in bone marrow derived macrophages, they identified using RNA-seq data the inward rectifying potassium current channel Kir2.1. They found this channel to be the regulator of resting membrane potential. They also used ML13 to block Kir2.1. But how they determined the specificity of this drug in blocked only this channel was not described.

They showed that ML133 suppressed a cluster of inflammatory genes as well as canonical genes such as IL-1beta and IL-1alpha mRNA. Increasing extracellular potassium and adding gramicidin showed similar suppression of the same genes, thus it appears that Kir 2.1 is an important regulator of the macrophage membrane potential and stimulates inflammatory macrophages in vitro through regulation of Kir 2 nutrient acquisition to feed the metabolic demands of macrophages during inflammation. Also repression of Kir 2.1 induced depletion fueled histone methylation marks indicating its role in epigenetic modification of macrophages.

The intent of these studies is very interesting since they deal with nutrient import in macrophages as being essential for the host defense of these cells. The central premise is notable is that ionic potassium efflux and plasma membrane potential control of macrophage nutrient acquisition, which is essential for normal antibacterial function of macrophages through expression of various gene programs and activation of glycolysis.

Much of the Discussion centers around the strategy of starving cancer cells and showing its applicability to inflammatory diseases. However there is no direct evidence to prove both conditions use the same pathways. As macrophages are highly plastic it's conceivable that processes are different in the two conditions. This speculative discussion should be considerably reduced.

While the focus is on nutrient acquisition there should also be discussion about other factors in mediating adaptive changes in macrophages such as activation of efferocytosis and phagocytosis which may be as important as changes in the membrane potential and activation of specific metabolic pathways. The authors do make a strong case for changes in macrophage membrane potential, other macrophage dependent anti-inflammatory mechanisms should also be discussed. The studies were made using bone marrow derived macrophages. Do specific tissue macrophages respond in a similar manner? Or are the results specific to immature bone marrow derived macrophages

What if any alterations in mitochondria change the phenotype of macrophages? The focus is solely in glycolytic pathway changes. There should be mention made why the mitochondrial is not involved.

When macrophages change their phenotype a key question is whether this is a reversible process. Do the cells ever come back to their baseline state and if so is this associated with normalization of metabolic functions of macrophages?

REVIEWERS' COMMENTS

Reviewer #5 (Remarks to the Author):

The paper explores the effects of changing membrane potential on inflammatory function of macrophages. Studies are made using LPS stimulated peritoneal macrophages or bone marrow macrophages in which membrane potential was measured by patch clamping. Increasing extracellular potassium ion concentration or applying gramicidin that elevated extracellular potassium resulted in membrane depolarization, promoted nutrient uptake which in turn activated glycolytic, pentose phosphate pathway, citric cycle, purine synthesis, phospholipid biosynthesis, amino acid metabolism, and biosynthesis. These results support the notion that PM depolarization promotes nutrient acquisition and subsequent anabolic metabolism and makes macrophages better host defense cells.

We appreciate the reviewer's comments.

Of the 54 annotated potassium channel genes in bone marrow derived macrophages, they identified using RNA-seq data the inward rectifying potassium current channel Kir2.1. They found this channel to be the regulator of resting membrane potential. They also used ML13 to block Kir2.1. But how they determined the specificity of this drug in blocked only this channel was not described.

The specificity of ML133 on Kir2.1 has been intensively assessed in the original studies discovering this inhibitor. ML133 was reported to exhibit greater than 100-fold selectivity for block of Kir2.1 compared with block of Kir1.1, and 17–40-fold selectivity against Kir7.1 and Kir4.1 channels (PMID: 21615117, 21433384). According to the product description from Selleckchem, ML133 is also recognized as a selective inhibitor for Kir2.1 (<https://www.selleckchem.com/products/ML-133-hcl.html>). A large number of literatures used ML133 to block Kir2.1 channel in various cell types with the concentrations from 10uM to 50 uM similar to those used in our study (e.g. PMID: 21615117, 21433384, 26627720, 26840527, 32220118, 26324774, 26029054, 27009332, 25472961, 27598576).

Second, according to the expression profile as listed in Figure 2A, to rule out possible off-target effect of ML133, we transfected another two K⁺ channels that are also expressed in macrophages including Kir4.1 and K_{Ca}3.1 in HEK293 cells. The results showed that 50uM ML133 had little effect on their currents as indicated below.

Last, we found that the degree of V_m depolarization by 25uM ML133 (from -42.85 mV to -30.6 mV, decrease by 28.6%) was similar to that by Kir2.1 deficiency (from -42.85 mV to -29.73 mV, decrease by 30.6%), which indicates that 25uM ML133 and Kir2.1 deficiency have comparable V_m depolarizing effects. Most importantly, no additional effect of ML133 in Kir2.1-deficient macrophages which further confirmed its specific effect on Kir2.1.

They showed that ML133 suppressed a cluster of inflammatory genes as well as canonical genes such as IL-1beta and IL-1alpha mRNA. Increasing extracellular potassium and adding gramicidin showed similar suppression of the same genes, thus it appears that Kir 2.1 is an important regulator of the macrophage membrane potential and stimulates inflammatory macrophages in vitro through regulation of Kir 2 nutrient acquisition to feed the metabolic demands of macrophages during inflammation. Also repression of Kir 2.1 induced depletion fueled histone methylation marks indicating its role in epigenetic modification of macrophages. The intent of these studies is very interesting since they deal with nutrient import in macrophages as being essential for the host defense of these cells. The central premise is notable is that ionic potassium efflux and plasma membrane potential control of macrophage nutrient acquisition, which is essential for normal antibacterial function of macrophages through expression of various gene programs and activation of glycolysis.

We appreciate these comments.

Much of the Discussion centers around the strategy of starving cancer cells and showing its applicability to inflammatory diseases. However there is no direct evidence to prove both conditions use the same pathways. As macrophages are highly plastic its conceivable that processes are different in the two conditions. This speculative discussion should be considerably reduced.

We agree with the reviewer's suggestion and have reduced the speculative discussion regarding to the situation of cancer in the manuscript.

While the focus is on nutrient acquisition there should also be discussion about other factors in mediating adaptive changes in macrophages such as activation of efferocytosis and phagocytosis which may be as important as changes in the membrane potential and activation of specific metabolic pathways. The authors do make a strong case for changes in macrophage membrane potential, other macrophage dependent anti-inflammatory mechanisms should also be discussed.

We do agree with the reviewer's opinion about the importance of other factors in mediating adaptive changes in macrophages. Except for inflammatory functions, efferocytosis and phagocytosis is also the important functions of macrophages. As showed by our data and other literatures that membrane potential depolarization triggered membrane PIP2 clustering and the subsequent endocytosis, it is likely that efferocytosis and phagocytosis may also be involved in the membrane dynamics. Metabolic reprogramming resulting from efferocytosis triggers activation of resolution pathways. Efferocytosis, besides its metabolic input, directly regulates macrophage polarization toward an anti-inflammatory and pro-homeostatic phenotype (PMID: 33927896, 30647530). Overall, although this study explored the most classical function of secreting inflammatory cytokines, more macrophage functions regulated by membrane potential under different physiological and pathological contexts are worthy to be further investigated.

Except for membrane potential depolarization that regulated the functions of macrophage, many mechanisms underlying the inflammatory regulation have been explored to date. For example, signaling molecules such as E3 ubiquitin ligase and phosphates could attenuate the transduction of inflammatory signaling pathways. Whether the membrane potential-mediated metabolic reprogramming and membrane phospholipid dynamics could affect these regulatory molecules activity (e.g. subcellular re-localization or attachment to different organelle membranes) is also interesting question for future exploration.

The studies were made using bone marrow drive macrophages. Do specific tissue macrophages respond in a similar manner? Or are the results specific to immature bone marrow derived macrophages

In this study, we tested bone marrow drive macrophages (BMDMs), mouse

peritoneal macrophages and macrophage cell lines, and all these cells showed similar results. Although possibly beyond the current scope, it is indeed interesting to examine whether different tissue resident macrophages such as Kupffer cells or microglia have similar (or tissue-specific) responses to the regulation of membrane potential. Given to the distinct microenvironment properties of different organs and tissues, it is highly worthy to investigate this mechanism under these tissue-specific contexts both physiologically and pathologically.

What if any alterations in mitochondria change the phenotype of macrophages? The focus is solely in glycolytic pathway changes. There should be mention made why the mitochondrial is not involved.

We appreciate the reviewer's suggestion about mitochondria. Mitochondria are critical for regulation of the activation, differentiation, and survival of macrophages. In response to various extracellular signals, such as microbial or viral infection, changes to mitochondrial metabolism and physiology could underlie the corresponding state of macrophage activation. These changes include alterations of oxidative metabolism, mitochondrial membrane potential, and tricarboxylic acid (TCA) cycling, as well as the release of mitochondrial reactive oxygen species (mtROS) and mitochondrial DNA (mtDNA) and transformation of the mitochondrial ultrastructure (PMID:34157289).

LPS stimulation greatly enhances macrophage glycolysis. ECAR of macrophages was upregulated, while OCR was mildly influenced upon LPS stimulation (PMID: 32937100, 30773464). Consistently, we also found changes in the glycolytic pathway, glycolytic offshoots including pentose phosphate pathway (PPP) and serine synthesis pathway (SSP). In addition, the mitochondrial folate cycle was also affected. Therefore, although the mitochondrial OXPHOS might be limitedly affected, the serine and folate metabolism in the mitochondria as well as its downstream SAM generation were involved in this regulation of Kir2.1.

When macrophages change their phenotype a key question is whether this is a reversible process. Do the cells ever come back to their baseline state and if so is this associated with normalization of metabolic functions of macrophages?

Diversity and plasticity are hallmarks of macrophages. Macrophage differentiation is highly dynamic. Responding to different microenvironmental cues, macrophages can be activated or de-activated to the resting state, or switch from one phenotype to the other. Accumulating evidences have shown that the spatial-temporal regulation of macrophage activation at different stages during inflammation and tissue repair. LPS normally polarizes macrophages to an M1-like inflammatory phenotype, but repeated LPS stimulation induces tolerance (PMID: 27427981). LPS-tolerant macrophages have an M2-like phenotype that is highly associated with immunosuppression. During the response to LPS and interferon- γ stimulation, metabolic reprogramming in macrophages is also highly dynamic. Specifically, the TCA cycle undergoes a two-stage remodeling: the early stage is characterized by a transient accumulation of intermediates including succinate and itaconate, whereas the late stage is marked by the subsidence of these metabolites. For persisting LPS stimulation, many metabolites

showed drastic differences in early and late phases of stimulation. It is particularly intriguing that several important metabolites in, or immediately derived from, the TCA cycle, including citrate, succinate, succinyl-CoA and itaconate, showed dynamic patterns similar to the production of the cytokines—a profound increase after 6–24h of stimulation followed by a reduction to levels similar to, or below, that in unstimulated macrophages or so-called baseline state (PMID: 32259027). So, the metabolism also adapts accordingly, and make the metabolic states consistent with the polarization state and functional phenotype of macrophages.